# The Cac1 subunit of histone chaperone CAF-1 organizes CAF-1-H3/H4 architecture and tetramerizes histones

Wallace H Liu[1], Sarah C Roemer[1], Yeyun Zhou[1], Zih-Jie Shen[2], Briana K Dennehey[3], Jeremy L Balsbaugh[4,5], Jennifer C Liddle[4,5], Travis Nemkov[6], Natalie G Ahn[4,5], Kirk C Hansen[6], Jessica K Tyler[2,3]*, Mair EA Churchill[1,6]*

[1]Department of Pharmacology, University of Colorado School of Medicine, Aurora, United States; [2]Department of Pathology and Laboratory Medicine, Weill Cornell Medicine, New York, United States; [3]Department of Epigenetics and Molecular Carcinogenesis, MD Anderson Cancer Center, Houston, United States; [4]Department of Chemistry and Biochemistry, University of Colorado, Boulder, Boulder, United States; [5]BioFrontiers Institute, University of Colorado, Boulder, Boulder, United States; [6]Program in Structural Biology and Biochemistry, University of Colorado School of Medicine, Aurora, United States

**Abstract** The histone chaperone Chromatin Assembly Factor 1 (CAF-1) deposits tetrameric (H3/H4)$_2$ histones onto newly-synthesized DNA during DNA replication. To understand the mechanism of the tri-subunit CAF-1 complex in this process, we investigated the protein-protein interactions within the CAF-1-H3/H4 architecture using biophysical and biochemical approaches. Hydrogen/deuterium exchange and chemical cross-linking coupled to mass spectrometry reveal interactions that are essential for CAF-1 function in budding yeast, and importantly indicate that the Cac1 subunit functions as a scaffold within the CAF-1-H3/H4 complex. Cac1 alone not only binds H3/H4 with high affinity, but also promotes histone tetramerization independent of the other subunits. Moreover, we identify a minimal region in the C-terminus of Cac1, including the structured winged helix domain and glutamate/aspartate-rich domain, which is sufficient to induce (H3/H4)$_2$ tetramerization. These findings reveal a key role of Cac1 in histone tetramerization, providing a new model for CAF-1-H3/H4 architecture and function during eukaryotic replication.

*For correspondence: jet2021@med.cornell.edu (JKT); Mair.Churchill@ucdenver.edu (MEAC)

## Introduction

Eukaryotic DNA is incorporated into chromatin structure via nucleosomes, which are units of ~146 DNA base pairs surrounding an octamer of histone proteins (*Kornberg, 1974*). The assembly and disassembly of nucleosomes is critical during DNA replication, as duplication of the genome requires concerted duplication of the resident histones. To facilitate this stepwise process (*Smith and Stillman, 1991*), eukaryotes employ histone chaperones, proteins that associate tightly with histones prior to histone deposition on DNA (reviewed in [*Ransom et al., 2010*; *Das et al., 2010*; *Burgess and Zhang, 2013*; *Gurard-Levin et al., 2014*]).

The CAF-1 (Chromatin Assembly Factor (1) histone chaperone is intimately engaged in replication-dependent nucleosome assembly of nascent (*Verreault et al., 1996*; *Smith and Stillman, 1989*; *Tyler et al., 2001*) and parental histones H3/H4, by way of recruitment to the replication fork through PCNA (Proliferating cell nuclear antigen) (*Shibahara and Stillman, 1999*; *Krawitz et al., 2002*; *Moggs et al., 2000*; *Rolef Ben-Shahar et al., 2009*). At the fork, parental H3/H4 histones are

**eLife digest** The DNA of a human, yeast or other eukaryotic cell is bound to proteins called histones to form repeating units called nucleosomes. Every time a eukaryotic cell divides, it must duplicate its DNA. Old histones are first removed from the nucleosomes before being re-assembled onto the newly duplicated DNA along with new histone proteins, producing a full complement of nucleosomes.

A group of proteins called the chromatin assembly factor 1 (or CAF-1 for short) helps to assemble the histones onto the DNA. CAF-1 is made up of three proteins, and binds to two copies of each of the histones known as H3 and H4. These are the first histones to be assembled onto the nucleosomes. It was not clear how the components of CAF-1 are organized, or how CAF-1 recognizes histones.

Liu et al. have now investigated the structure of CAF-1 and its interactions with the H3 and H4 histones by studying yeast proteins and cells. Yeast is a good model system because yeast CAF-1 is smaller and easier to isolate than human CAF-1, yet still performs the same essential activities. Using a combination of biochemical and biophysical techniques, Liu et al. found that one of the three proteins that makes up yeast CAF-1 – called Cac1 – forms a scaffold that supports the other CAF-1 proteins and histones H3 and H4. Moreover, a specific part of Cac1 is able to bind to these histones and assemble two copies of each of them to prepare for efficient nucleosome assembly.

Further experiments revealed the specific areas where the CAF-1 proteins interact with each other and with the histones, determined how strong those interactions are, and confirmed that these interactions play important roles in yeast.

Overall, the results presented by Liu et al. provide new insights into the structure of CAF-1 bound to H3 and H4. In order to understand in detail how CAF-1 helps to assemble histones onto DNA, future work needs to capture three-dimensional snapshots of the different steps in this process. Further investigation is also needed to discover how CAF-1 cooperates with other factors that promote DNA duplication.

inherited in a conservative manner, in which intact $(H3/H4)_2$ tetramers are partitioned onto the replicated daughter strands (*Prior et al., 1980*; *Xu et al., 2010*). This model predicts CAF-1 will associate with $(H3/H4)_2$ tetramers, rather than a single H3/H4 dimer. Multiple biophysical studies confirm that the monomeric form of CAF-1 tetramerizes H3/H4 (*Winkler et al., 2012b*; *Liu et al., 2012*). Other monomeric chaperones such as Asf1 bind one H3/H4 dimer (*English et al., 2005*, *2006*), whereas those that accommodate $(H3/H4)_2$ tetramers – including Nap1, Vps75, Mcm2 and Rtt106 – form homodimers themselves (*Bowman et al., 2011*; *Su et al., 2012*; *Hammond et al., 2016*; *Huang et al., 2015*; *Richet et al., 2015*). Together, the evidence suggests that CAF-1 induces a unique mechanism of H3/H4 oligomerization.

The budding yeast CAF-1 complex consists of subunits Cac1, Cac2, and Cac3 (*Figure 1A*), which differ in their ability to bind H3/H4. GST-tagged Cac1, but not the other subunits, can co-immunoprecipitate endogenous human histones when expressed in HeLa cells (*Li et al., 2008*). A single lysine to arginine substitution on H3 residue 56 abolished Cac1 binding, implicating the first alpha helix (αN) in this interaction. Biophysical studies with the other subunits show that the *H. sapiens* and *D. melanogaster* homologs of Cac3 can bind H3/H4 with varying affinities (*Song et al., 2008*; *Murzina et al., 2008*; *Nowak et al., 2011*; *Schmitges et al., 2011*). However, dissociation constants for Cac1 and Cac2 binding to H3/H4 have not been reported, and neither have the subunit interactions required to induce the 1:2 chaperone:histone dimer stoichiometry.

The architecture of the CAF-1 and CAF-1-H3/H4 complexes is not known in detail. Some biochemical evidence points to specific interactions between Cac1 and Cac2, and between Cac1 and Cac3. Substitution of Cac1 leucine 276 to proline blocks Cac3 binding (*Krawitz et al., 2002*), and GST-tagged Cac1 residues 215–429 are sufficient to bind Cac3 in vitro. In addition, the last one-third of the Cac1 human homolog, p150, is sufficient to bind Cac2 (*Kaufman et al., 1995*). Whether these interactions are critical for H3/H4 binding and oligomerization is also unknown.

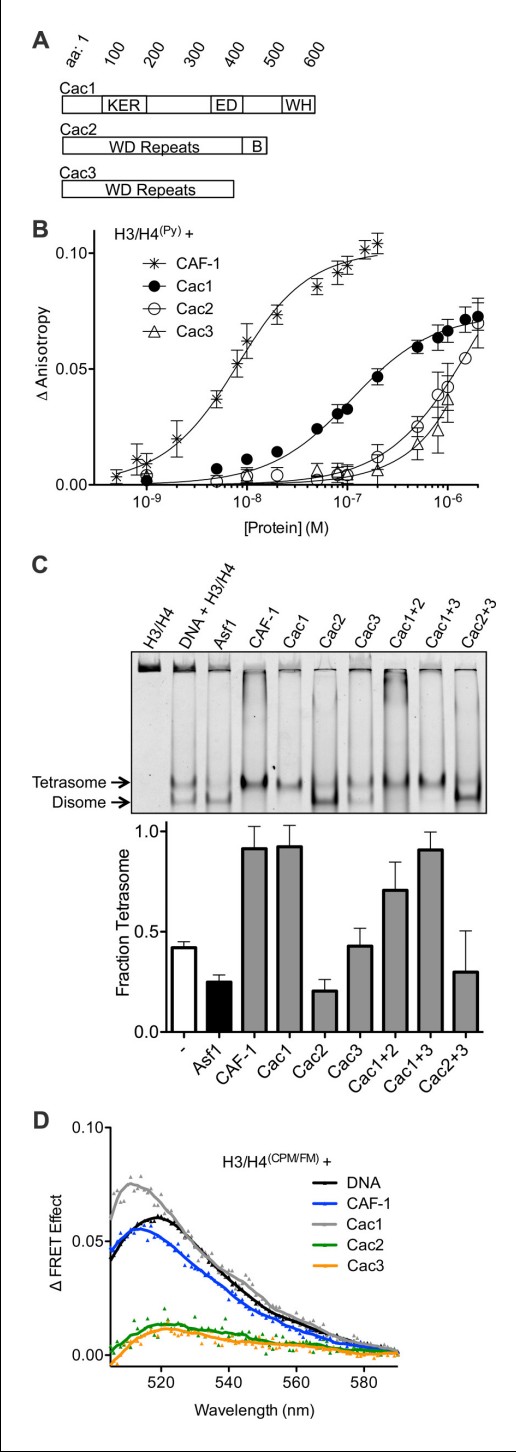

**Figure 1.** The Cac1 subunit is sufficient for (H3/H4)$_2$ tetramerization. (**A**) Schematic of domains in the individual CAF-1 subunits. (**B**) H3/H4$^{(Py)}$ binding to individual CAF-1 subunits. Fluorescence anisotropy of 25 nM pyrene-labeled H3/H4 was monitored with titration of individually purified Cac1, Cac2, or Cac3 in Histone Buffer (H.B.: 20 mM Tris, 150 mM KCl, 2 mM MgCl$_2$, 0.5 mM TCEP, 1% Glycerol, 0.05% BRIJ-35.) The CAF-1 complex was titrated into 5 nM H3/H4$^{(Py)}$. The

In this study, we investigate the CAF-1 and CAF-1-H3/H4 complexes using biophysical, structural, and functional validation approaches to gain insight into the mechanism of CAF-1 subunit interactions that promote H3/H4 tetramerization. The data indicate that Cac1 is the major subunit required and is sufficient for binding and tetramerizing H3/H4. Mass spectrometric analyses using both hydrogen/deuterium exchange and chemical cross-linking reveal H3/H4-induced conformational changes, as well as extensive cross-links between the C-terminus of Cac1 with H3 and H4. Although structural and biophysical analyses reveal a homo-dimerization motif at the Cac1 C-terminus, this is not sufficient for H3/H4 tetramerization. Rather, the Cac1 glutamate/aspartate (ED) domain is also required to tetramerize H3/H4 in vitro. These results support an architectural model for the replication-coupled CAF-1-H3/H4 complex, in which Cac1 scaffolds all protein-protein interactions and is predominantly involved in H3/H4 tetramerization.

## Results

### The Cac1 subunit binds to H3/H4 and is sufficient to assemble (H3/H4)$_2$ tetramers

To measure the contribution of each individual CAF-1 subunit in histone binding, the individual subunits were purified for use in fluorescence anisotropy experiments with pyrene-labeled H3/H4 (H3/H4$^{(Py)}$) (*Figure 1A,B*, *Figure 1—figure supplement 1*). Each subunit was titrated independently into a fixed concentration of H3/H4$^{(Py)}$, inducing a concentration-dependent increase in pyrene fluorescence anisotropy that enabled binding constants to be measured (*Table 1*). Consistent with prior experiments, CAF-1 binds to H3/H4$^{(Py)}$ with low nanomolar affinity (K$_{Dapp}$ = 5.3 nM) (*Winkler et al., 2012b*; *Liu et al., 2012*). Although all CAF-1 subunits increased the anisotropy of H3/H4$^{(Py)}$ at high (1 µM) concentrations, only Cac1 achieved saturable binding with submicromolar affinity for H3/H4 (K$_{Dapp}$ = 97 nM). Cac2 and Cac3 titrations, on the other hand, did not reach saturation and displayed weak affinity for H3/H4. These results suggest that Cac1 contributes substantially to H3/H4 binding, and the other subunits provide accessory interactions to promote a high-affinity CAF-1-H3/H4 complex.

Recent reports showed that CAF-1 not only binds to H3/H4, but arranges the histones into a nucleosomal (H3/H4)$_2$ tetramer (*Winkler et al., 2012b*; *Liu et al., 2012*). Since all CAF-1 subunits

*Figure 1 continued*

curves were fitted using *Equation 3*. (C) A representative EMSA separating histone:DNA species as disomes or tetrasomes. 1.6 µM of the indicated histone chaperone or CAF-1 subunit was incubated with 0.2 µM H3/H4$^{(FM)}$ dimer, prior to addition of 0.4 µM 80 bp DNA. The bar graph shows the mean and standard deviation of fraction of tetrasomes formed, calculated by *Equation 5*, from at least three independent experiments. (D) FRET of mixed labeled H3/H4$^{(CPM/FM)}$. Spectra were obtained for 10 nM of labeled histones incubated with 0.2 µM CAF-1 or DNA, or 1 µM CAF-1 subunit. The FRET Effect was calculated using *Equation 4* from at least three independent experiments.

The following figure supplement is available for figure 1:

**Figure supplement 1.** Purified proteins used in this study.

bind H3/H4, albeit very weakly for Cac2 and Cac3, potentially one or more of these subunits in combination might be required to form tetrasomes ((H3/H4)$_2$ tetramer bound to DNA). To test this possibility, we utilized an EMSA capable of resolving different stoichiometric species of fluorescein-labeled H3/H4 (H3/H4$^{(FM)}$) bound to 80 base pair nucleosome-positioning DNA (*Donham et al., 2011*). Previously, this method revealed that Asf1 favors deposition of H3/H4$^{(FM)}$ as a disome species, which consists of an H3/H4 dimer bound to DNA (*Donham et al., 2011*; *Scorgie et al., 2012*), whereas CAF-1 favors formation of tetrasomes (*Liu et al., 2012*). Surprisingly, Cac1 alone induces tetrasome formation (*Figure 1C*). In contrast, Cac3 does not alter the basal fraction of tetrasomes formed. Cac2 induces more disome species, but the pre-formed Cac1/Cac2 complex deposits tetrasomes, suggesting that the Cac1 subunit promotes tetrasome assembly.

To confirm that Cac1-induced tetrasome formation was a direct result of (H3/H4)$_2$ tetramerization independent of DNA, we used a mixed-fluorophore Förster resonance energy transfer (FRET) assay with equimolar concentrations of fluorescein (FM)-labeled and 7-Diethylamino-3-(4'-Maleimidylphenyl)-4-Methylcoumarin (CPM)-labeled H3/H4 dimers (H3/H4$^{(CPM/FM)}$). As we previously reported, H3/H4$^{(CPM/FM)}$ exhibits CPM to FM FRET in the presence of DNA or CAF-1 (*Figure 1D*), but not in the presence of Asf1, which binds to a dimer of H3/H4 (*Liu et al., 2012*). Neither Cac2 nor Cac3 induced FRET, but Cac1 increased the FRET signal of H3/H4$^{(CPM/FM)}$ to a similar extent as CAF-1 or DNA. Together, these data reveal that Cac1 is sufficient for (H3/H4)$_2$ tetramerization.

## H3/H4 binding to CAF-1 promotes changes in hydrogen/deuterium exchange

To elucidate potential interaction regions between CAF-1 and H3/H4, we subjected the CAF-1 and CAF-1-H3/H4 complexes to hydrogen/deuterium exchange (HX) at 10°C, followed by liquid chromatography-mass spectrometry (LC-MS/MS) analysis (*Hoofnagle et al., 2003*). Hydrogen exchange was carried out for 30 and 60 min, followed by acid quench, pepsin digestion and mass analysis. The parent spectra (MS1) were used to compute the number of deuterons incorporated per peptide,

**Table 1.** $K_D$ values from pyrene fluorescence anisotropy of CAF-1 subunits and H3/H4.

| Pyrene-labeled protein | Binding partner | $K_D$ or $K_{Dapp}$ (M) |
|---|---|---|
| H3/H4 | CAF-1 | $5.3 \pm 0.9 \times 10^{-9}$ |
| H3/H4 | Cac1 | $9.7 \pm 1.8 \times 10^{-8}$ |
| H3/H4 | Cac2 | n.c. |
| H3/H4 | Cac3 | n.c. |
| Cac1$^{386}$ | Cac1$^{386}$ | $2.6 \pm 0.2 \times 10^{-8}$ |
| Cac1$^{454}$ | Cac1$^{454}$ | $2.5 \pm 0.2 \times 10^{-8}$ |
| Cac1$^{386}$ | H3/H4 | $2.1 \pm 0.5 \times 10^{-7}$ |
| Cac1$^{454}$ | H3/H4 | n.c. |
| Cac1$^{386}$ | Cac2 | $1.3 \pm 0.4 \times 10^{-6}$ |

n.c. not calculated

and the difference in deuteron uptake between samples was used to calculate changes in HX between bound and unbound states (*Figure 2A*; *Supplementary file 1A*). Without available structural data for the CAF-1 proteins, we used the PHYRE2 server (*Kelley et al., 2015*) to obtain high-scoring Cac2 and Cac3 homology models for interpretation of HX (*Figure 2B*).

Several regions of Cac1 show changes in HX upon binding H3/H4. In forming the CAF-1-H3/H4 complex, the C-terminal half of Cac1 showed both increased HX (amino acids 550–591) and decreased HX (amino acids 304–322, 340–360, and 463–473) (*Figure 2A*, *Figure 2—figure supplement 1*). The regions with decreased HX were candidates for potential protein-protein interactions between Cac1 and H3/H4 (*Percy et al., 2012*).

To evaluate the significance of these HX changes at the Cac1 C-terminus in vivo, we examined the effect of deleting the corresponding amino acids on CAF-1 function in *Saccharomyces cerevisiae*. Using yeast deleted for *CAC1,* we introduced either empty vector, a vector expressing wild type Cac1 or mutant Cac1 with a series of deletions and substitutions (*Tables 2* and *3*; *Supplementary file 1B*). CAF-1 is known to be required for resistance to DNA damaging agents that cause DNA double-strand breaks (*Linger and Tyler, 2005*), due to its role in chromatin assembly during DNA repair. Yeast lacking Rad52 were included to provide a control showing extreme sensitivity to DNA double-strand damage. Rad52 is essential for homologous recombination, the central repair mechanism used in yeast. Whereas *cac1* yeast were sensitive to the radiomimetic zeocin, re-introduction of the vector expressing wild type *CAC1* rescued cells from DNA damage (*Figure 2C*). Of the strains that expressed Cac1 mutant proteins, the DNA damage resistance afforded by Cac1 was noticeably compromised by the 304–322, 463–473 and 578–580 deletions in Cac1 (*Figure 2C*). These deletions did not significantly affect the expression levels of the Cac1 protein (*Table 3* and data not shown). We conclude that the decreased HX in these regions upon H3/H4 binding likely reflects their importance for the physiological complex.

Several predicted loops in Cac2 also show changes in HX upon H3/H4 association. The Cac2 subunit is confidently homology-modeled as a WD-repeat β propeller structure, with an intrinsically disordered C-terminus (not shown). The HX data reveal that the N-terminal loop and propeller blades 5 and 6 are protected from exchange following H3/H4 binding (*Figure 2A and B*). To determine whether these amino acids also influence the physiological response to DNA damage in yeast, we designed mutations (*Tables 2* and *3*; and *Supplementary file 1B*) in the loop regions, including the N-terminal loop (amino acids 7–17), a loop in blade 5 (amino acids 206–207), two loops in blade 6 (amino acids 273/275/276/277 and 316/318), and the disordered C-terminus (amino acids 425–468), in order to minimize disruption of the Cac2 structure. Vectors expressing wild type Cac2 or Cac2 bearing mutations within these regions were introduced into the *cac2* strain. Only the Δ1–15, V273A/P275A/S276A/G277A, L316E/L318E, and Δ425–468 mutations displayed various degrees of sensitivity to zeocin-induced DNA damage (*Figure 2C*). These mutations did not significantly alter the expression levels of Cac2. From these data, we conclude that Cac2 regions at the N-terminus and in blade 6 identified in the HX analysis are functionally important in vivo.

Cac3 exhibits little change in HX with H3/H4 association. Cac3 was modeled from the human homolog RbAp46, which is also a WD-repeat β-propeller structure. This Cac3 model also contains an N-terminal alpha helix, which is used by RbAp46 for binding the N-terminal helix of histone H4 (*Murzina et al., 2008*). The HX data show no significant changes in deuteration in Cac3 when the CAF-1 complex is compared to CAF-1-H3/H4 (*Figure 2A and B*). Taken together, the HX and in vivo results indicate that H3/H4 binding to CAF-1 primarily promotes HX changes to Cac1 and Cac2, but not Cac3, in regions that are important for CAF-1 function in vivo.

## The C-terminal half of Cac1 directly cross-links to tail and core residues in H3 and H4

Cac1 is sufficient to tetramerize H3/H4 and also appears to act as a scaffold for the other CAF-1 subunits. However, the HX experiment reports Cac1 regions with deuteration changes that could be due either to direct protein-protein interactions, or indirect effects resulting from allosteric conformational changes or conformational dynamics. To provide more direct information about the physical interactions that shape CAF-1-H3/H4 architecture, we used chemical cross-linking with DSS (disuccinimidyl suberate) or EDC (1-ethyl-3-(3-dimethylaminopropyl) carbodiimide hydrochloride) coupled to mass spectrometry (CX-MS) (*Walzthoeni et al., 2013*). Identification of linked residues

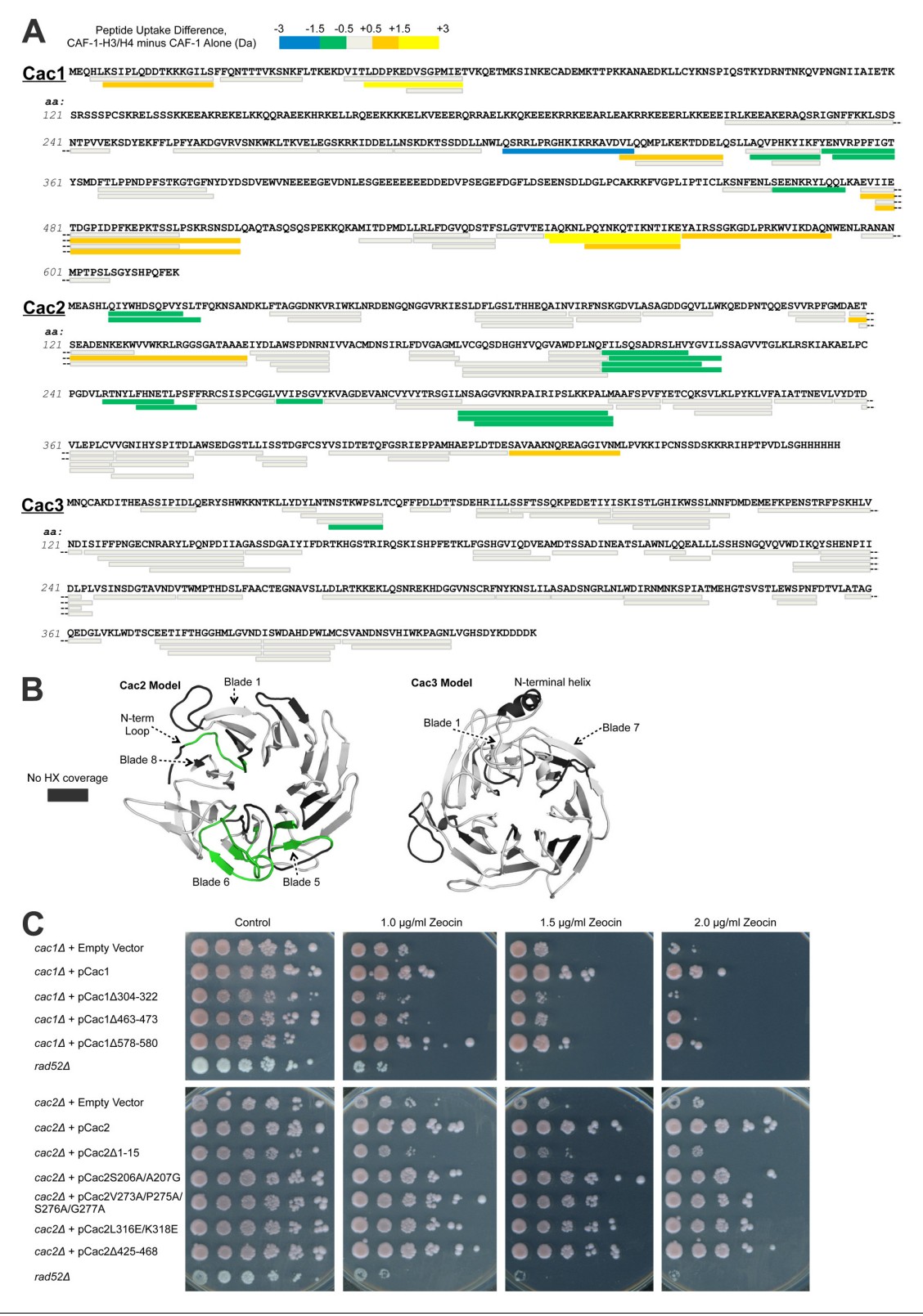

**Figure 2.** Hydrogen/deuterium exchange of CAF-1, and CAF-1-H3/H4 complexes. (**A**) The sequences of the three individual CAF-1 subunits are shown. Each bar represents an individual identical peptide observed in the protein between the compared samples, plotted as the difference in deuteron uptake between the CAF-1 and CAF-1-H3/H4 samples (i.e., difference = CAF-1-H3/H4 – CAF-1 only). The differences in deuteron uptake at 60' are colored according to the legend. The 'cooler' colors (green, blue, and purple) represent an increase in apparent protection for the peptide in CAF-1-

*Figure 2 continued on next page*

*Figure 2 continued*

H3/H4 compared to the CAF-1 sample, whereas the 'warmer' colors (orange, yellow, and red) represent decreased apparent protection. Peptide coverage was approximately 60%, 80%, and 80% for Cac1, Cac2 and Cac3, respectively. (B) Differences in HX at 60' were mapped on PHYRE2 models of Cac2 and Cac3. The coloring scheme is the same as for A, but amino acids with no coverage are colored dark gray to distinguish these residues from those that have coverage but did not exchange significantly. (C) The top panel shows five fold serial dilution analysis of strain CFY53 (*cac1*) with the vector pCac1 introduced that was either empty, expressed wild type Cac1 or Cac1 with the indicated amino acid changes. The bottom shows five fold serial dilution analysis of strain CFY54 (*cac2*) with the vector pCac2 introduced that was either empty, expressed wild type Cac2 or Cac2 with the indicated amino acid changes.

The following figure supplement is available for figure 2:

**Figure supplement 1.** Peptide coverage in HX-MS.

can indicate regions involved in protein-protein interactions, and possibly identify tertiary and quaternary interactions.

To identify cross-linked peptides, we digested the CAF-1 or CAF-1-H3/H4 complexes with trypsin, trypsin and GluC, or trypsin and LysC, followed by LC-MS/MS. We used the Protein Prospector package (*Chu et al., 2010*) from UCSF to identify cross-linked peptides. The collective data from the searches enabled us to create a map of linkages for the CAF-1 subunits and H3/H4 (*Figure 3A and B*; *Supplementary file 1C*). In addition to inter-subunit cross-links, many intra-protein cross-links were observed (*Figure 3*, *Figure 3—figure supplement 1*). Such cross-links are especially prevalent in Cac1 within the first fifty amino acids, residues 118–334, and residues 460–593. These may reveal folded regions in Cac1, since unfolded, flexible domains likely cannot make stable cross-links (*Leitner et al., 2010*).

Within the CAF-1 and CAF-1-H3/H4 complexes, cross-links were observed between subunits Cac1 and Cac2, Cac1 and Cac3, but not between Cac2 and Cac3. Cross-links between the Cac1 C-terminus to Cac2 (*Figure 3A and B*) are consistent with the last third of Cac1 binding to Cac2 (*Kaufman et al., 1995*). In addition, cross-links from Cac2 to the middle domain of Cac1 were also observed, suggesting that modular interactions scaffold Cac1-Cac2 binding.

The Cac1 cross-links to Cac3 (Cac1$^{K235}$ to Cac3$^{K287}$ and Cac1$^{K282}$ to Cac3$^{K307}$) (*Figure 3A and B*) complement a prior observation in which a Cac1 L276P mutation abrogates binding to Cac3 (*Krawitz et al., 2002*). To determine whether these Cac3 residues influence CAF-1 function in vivo, we deleted the *CAC3* gene in *S. cerevisiae*, and introduced either an empty vector, a vector expressing wild type Cac3 or Cac3 bearing deletions (Δ306–309 or Δ287–290) in predicted loops (*Figure 3C* and *Table 2*). Although wild type *CAC3* rescued the DNA damage phenotype resulting from zeocin treatment, the deletions both failed to do so. This result suggests that these Cac3 residues are important for CAF-1 structure and function.

Many cross-links were observed between CAF-1 and H3. Multiple residues within the Cac1 C-terminus (577–593) are proximal to both the H3 N-terminal tail lysines (4 and 18) and a C-terminal lysine 122. The predicted N-terminal helix of Cac3 also cross-links to the H3 N-terminus. Notably, Cac1 lysines 444 and 577 cross-link to the αN and α1 helices of H3 (Cac1$^{K444}$ to H3$^{K64}$, Cac1$^{K577}$ to H3$^{K64}$, and Cac1$^{K444}$ to H3$^{E59}$) (*Figure 3B*). These residues are close in space to H3-K56 and reinforce reports that mutation or acetylation of H3-K56 alters binding to Cac1 (*Winkler et al., 2012b*; *Li et al., 2008*).

The C-terminal half of Cac1 is also involved in cross-linking to histone H4. Cac1 residues 577–588 cross-link to the H4 N-terminal tail and to the core through H4$^{K79}$. K79 is immediately adjacent to the H4 α2 helix, which also cross-links to other Cac1 residues (H4$^{E53}$ to Cac1$^{K317}$ and H4$^{K59}$ to Cac1$^{E464}$) (*Figure 3B*). Importantly, the Cac1 residues involved in histone cross-linking are located within peptides that exhibit significant changes in HX upon H3/H4 binding (*Figure 2A*). Notably, these peptides include residues 317 and 464, the deletions of which compromise CAF-1-dependent resistance to DNA damage in vivo (*Figure 2C*). Thus, the C-terminal half of Cac1 is involved in cross-links to both the tail and core domains of H3 and H4.

**Table 2.** Yeast strains and plasmids.

| Strain | Mutation | Genotype | | Reference |
|---|---|---|---|---|
| w1588-4a | *WT* | *Mat alpha; leu2-3,112; ade2-1; can1-100; his3-11,15; ura3-1; trp1-1; RAD5* | | Gift from R. Rothstein |
| CFY53 | *cac1Δ* | *Mat alpha; leu2-3,112; ade2-1; can1-100; his3-11,15; ura3-1; trp1-1; RAD5 cac1Δ::NAT* | | This study |
| CFY54 | *cac2Δ* | *Mat alpha; leu2-3,112; ade2-1; can1-100; his3-11,15; ura3-1; trp1-1; RAD5 cac2Δ::NAT* | | This study |
| CFY58 | *cac3Δ* | *Mat alpha; leu2-3,112; ade2-1; can1-100; his3-11,15; ura3-1; trp1-1; RAD5 cac3Δ::NAT* | | This study |
| JKT004 | *rad52Δ* | *MAT a rad52::TRP1; trp1-1; ura3-1; can1-100; ADE; bar1::LEU2; his3-11; GAL* | | *Ramey et al. (2004)* |
| **Plasmid** | | **Characteristics** | | **Reference** |
| pRS315 (EV) | | CEN6 ARSH4 LEU2 | | *Sikorski and Hieter (1989)* |
| pCac1 | | pRS315-Cac1 | | This study |
| pCac2 | | pRS315-Cac3 | | This study |
| pCac3 | | pRS315-Cac3 | | This study |
| pCac1Δ233-237 | | pRS315-Cac1 aa 233-237 deleted | | This study |
| pCac1Δ280-284 | | pRS315-Cac1 aa 280 to 284 deleted | | This study |
| pCac1Δ304-322 | | pRS315-Cac1 aa 304 to 322 deleted | | This study |
| pCac1Δ340-360 | | pRS315-Cac1 aa 340-360 deleted | | This study |
| pCac1Δ428-432 | | pRS315-Cac1 aa 428-432 deleted | | This study |
| pCac1K442E/R443E/K444E | | pRS315-Cac1 with the mutation K442E/R443E/K444E | | This study |
| pCac1Δ463-473 | | pRS315-Cac1 aa 463 to 473 deleted | | This study |
| pCac1Δ497-501 | | pRS315-Cac1 aa 497 to 501 deleted | | This study |
| pCac1Δ574-584 | | pRS315-Cac1 aa 574-584 deleted | | This study |
| pCac1Δ578-580 | | pRS315-Cac1 aa 578 to 580 deleted | | This study |
| pCac1Δ576-606 | | pRS315-Cac1 aa 576-606 deleted | | This study |
| pCac1Δ578-580 | | pRS315-Cac1 aa 578 to 580 deleted | | This study |
| pCac2Δ1-15 | | pRS315-Cac2 aa 1 to 15 deleted | | This study |
| pCac2E70K | | pRS315-Cac2 with the mutation E70K | | This study |
| pCac2D91K/D92K | | pRS315-Cac2 with the mutation D91K/D92K | | This study |
| pCac2S206A/A207G | | pRS315-Cac2 with the mutation S206A/A207G | | This study |
| pCac2V273A/P275A/S276A/G277A | | pRS315-Cac2 with the mutation V273A/P275A/S276A/G277A | | This study |
| pCac2I274A/S276A | | pRS315-Cac2 with the mutation I274A/S276A | | This study |
| pCac2D248K/E285K | | pRS315-Cac2 with the mutation D248K/E285K | | This study |
| pCac2R295E | | pRS315-Cac2 with the mutation R295E | | This study |
| pCac2K306A/N307A/R308A | | pRS315-Cac2 with the mutation K306A/N307A/R308A | | This study |
| pCac2L316A/K318A | | pRS315-Cac2 with the mutation L316A/K318A | | This study |
| pCac2L316E/K318E | | pRS315-Cac2 with the mutation L316E/K318E | | This study |
| pCac2Δ371-373 | | pRS315-Cac2 aa 371 to 373 deleted | | This study |
| pCac2M417A/H418A/E420A | | pRS315-Cac2 with the mutation M417A/H418A/E420A | | This study |
| pCac2Δ425-468 | | pRS315-Cac2 aa 425-468 deleted | | This study |
| pCac2Δ445-468 | | pRS315-Cac2 aa 445-468 deleted | | This study |
| pCac2K447E/K448E | | pRS315-Cac2 with the mutation K447E/K448E | | This study |
| pCac3K284A/K285A/E286A | | pRS315-Cac3 with the mutation K284A/K285A/E286A | | This study |
| pCac3Δ306-309 | | pRS315-Cac3 deleted aa 306 to 309 | | This study |
| pCac3Δ287-290 | | pRS315-Cac3 deleted aa 287 to 290 | | This study |

**Table 3.** Yeast mutants and phenotypes observed.

| Mutant | Rationale for mutant | Zeocin resistance | Protein expression |
|---|---|---|---|
| Cac1Δ233-237 | Cross-link to Cac3 (*Figure 3B*) | Sensitive | No |
| Cac1Δ280-284 | Cross-link to Cac3 (*Figure 3A,B*) | Sensitive | No |
| Cac1Δ304-322 | HX change with H3/H4 and cross-link to Cac2 (*Figure 2A,3B*) | Very sensitive | Yes |
| Cac1Δ340-360 | HX change with H3/H4 (*Figure 2A*) | Sensitive | No |
| Cac1Δ428-432 | In ED-rich Region (*Figure 1A*) | Not sensitive | Yes |
| Cac1K442E/K443E/K444E | Cross-link to H3 (*Figure 3B*) | Not sensitive | Yes |
| Cac1Δ463-473 | HX change and cross-link to H3/H4 (*Figure 2A, 3B*) | Sensitive | Yes |
| Cac1Δ497-501 | Cross-link to Cac2 (*Figure 3B*) | Little sensitive | Yes |
| Cac1Δ574-584 | HX change and cross-link to H3/H4 (*Figure 2A, 3B*) | Little sensitive | Yes |
| Cac1Δ578-580 | HX change and cross-link to H3/H4 (*Figure 2A, 3B*) | Sensitive | Yes |
| Cac1Δ575-606 | HX change and cross-link to H3/H4 (*Figure 2A, 3B*) | Very sensitive | No |
| Cac2Δ1-15 | HX change with H3/H4 (*Figure 2A*) | Very sensitive | Yes |
| Cac2E70K | Loop next to Cac2 N-terminal loop (*Figure 2A*) | Not sensitive | Yes |
| Cac2D91K/D92K | Cross-link to Cac1 (*Figure 3A*) | Sensitive | Yes |
| Cac2S206A/A207G | HX change with H3/H4 (*Figure 2A*) | Sensitive | Yes |
| Cac2V273A/P275A/S276A/G277A | HX change with H3/H4 (*Figure 2A*) | Sensitive | Yes |
| Cac2I274A/S276A | HX change with H3/H4 (*Figure 2A*) | Not sensitive | Yes |
| Cac2D284K/E285K | Cross-link to Cac1 (*Figure 3A*) | Not sensitive | Yes |
| Cac2R295E | Loop between Cac2 blades 5 and 6 (*Figure 2A*) | Sensitive | No |
| Cac2K306A/N307A/R308A | HX change with H3/H4 (*Figure 2A*) | Not sensitive | Yes |
| Cac2L316A/K318A | HX change with H3/H4 (*Figure 2A*) | Sensitive | Yes |
| Cac2L316E/K318E | HX change with H3/H4 (*Figure 2A*) | Sensitive | Yes |
| Cac2Δ371-373 | Loop next to Cac2 N-term loop and blade 6 (*Figure 2A*) | Sensitive | Yes |
| Cac2M417/H418A/E420A | C-terminal loop in Cac2 | Not sensitive | Yes |
| Cac2Δ425-468 | HX change with H3/H4 (*Figure 2A*) | Not sensitive | Yes |
| Cac2Δ445-468 | C-terminal loop in Cac2 | Not sensitive | Yes |
| Cac3K284A/K285A/E286A | Cross-link to Cac1 (*Figure 3B*) | Not sensitive | Yes |
| Cac3Δ287-290 | Cross-link to Cac1 (*Figure 3B*) | Sensitive | Yes |
| Cac3Δ306-309 | Cross-link to Cac1 (*Figure 3A*) | Sensitive | Yes |

## The Cac1 C-terminal region is sufficient to promote (H3/H4)$_2$ tetramerization

Although chemical capture experiments generate useful models for protein-protein interactions, the presence or absence of captured cross-links, or abundance of cross-links, are not indicative of equilibrium binding constants, which should be measured through biophysical experiments. As the Cac1 C-terminus has extensive cross-links with H3/H4, we reasoned that this entire region might be responsible for H3/H4 binding and tetramerization. To test this hypothesis and identify such a minimal region, serial N-terminal deletions were designed, which included the ED domain through the C-terminus. Truncated proteins beginning at residues 386 (Cac1$^{386}$), 421 (Cac1$^{421}$), 454 (Cac1$^{454}$), and 457 (Cac1$^{457}$) were expressed and purified from *E. coli*. Cac1$^{386}$, Cac1$^{421}$, and Cac1$^{457}$ were compared for the ability to induce histone tetramerization in the H3/H4$^{(FM)}$ DNA deposition EMSA, as well as in the H3/H4$^{(CPM/FM)}$ FRET assay (*Figure 4A,B*, *Figure 4—figure supplement 1*). Both biophysical methods reveal that amino acids 386–606 within Cac1 are competent for histone tetramerization, whereas Cac1$^{421}$ and Cac1$^{457}$ are not. These results suggest that Cac1 residues 386–457 – which overlap much of the ED domain – may be responsible for a significant part of the binding

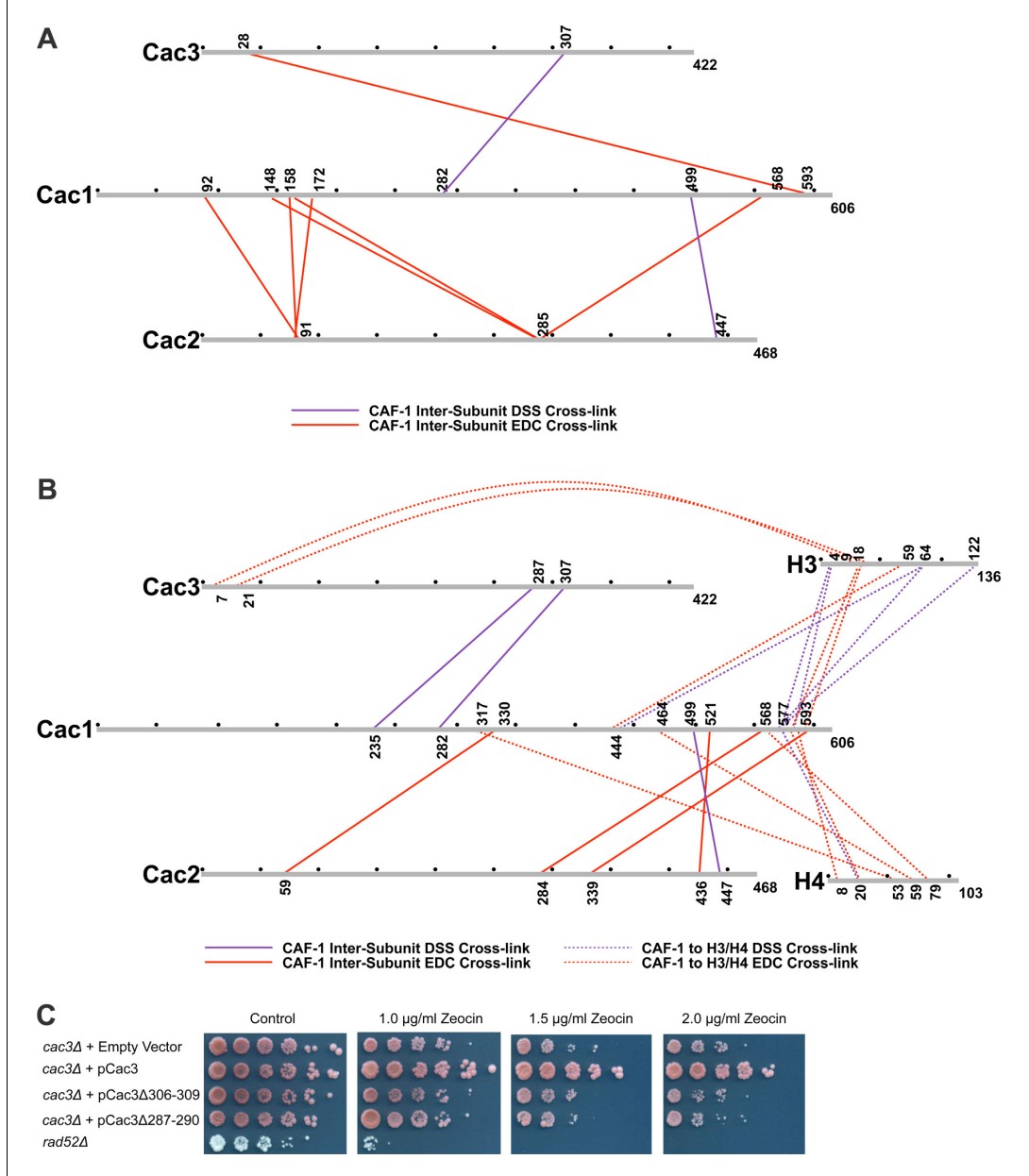

**Figure 3.** Chemical cross-linking of CAF-1 and CAF-1-H3/H4 complexes. (**A**) CAF-1 or (**B**) CAF-1-H3/H4 complexes were covalently cross-linked with DSS or EDC, then digested and run on an LTQ-Orbitrap. Cross-linked peptides were analyzed using Protein Prospector. The primary sequences are depicted in gray bars, with each gray circle marking 50 amino acid segments. DSS cross-links are shown in purple and EDC cross-links are in red. DSS leaves a 11.4 Å spacer arm between covalently-linked amine groups. EDC treatment results in a zero length cross-link between amine and carboxyl groups. The inter-subunit cross-links are represented as solid lines and cross-links to H3 and H4 are shown as dotted lines. (**C**) Analysis of Cac3 mutants in yeast. Cac3 mutants were subjected to zeocin-induced DNA damage response in vivo. The panel shows five fold serial dilution analysis of strain CFY58 (cac3) with the vector pCac3 introduced that was either empty (EV), expressed wild type Cac1, or Cac1 with the indicated amino acid changes.

The following figure supplement is available for figure 3:

**Figure supplement 1.** Intra-Cac1 cross-links.

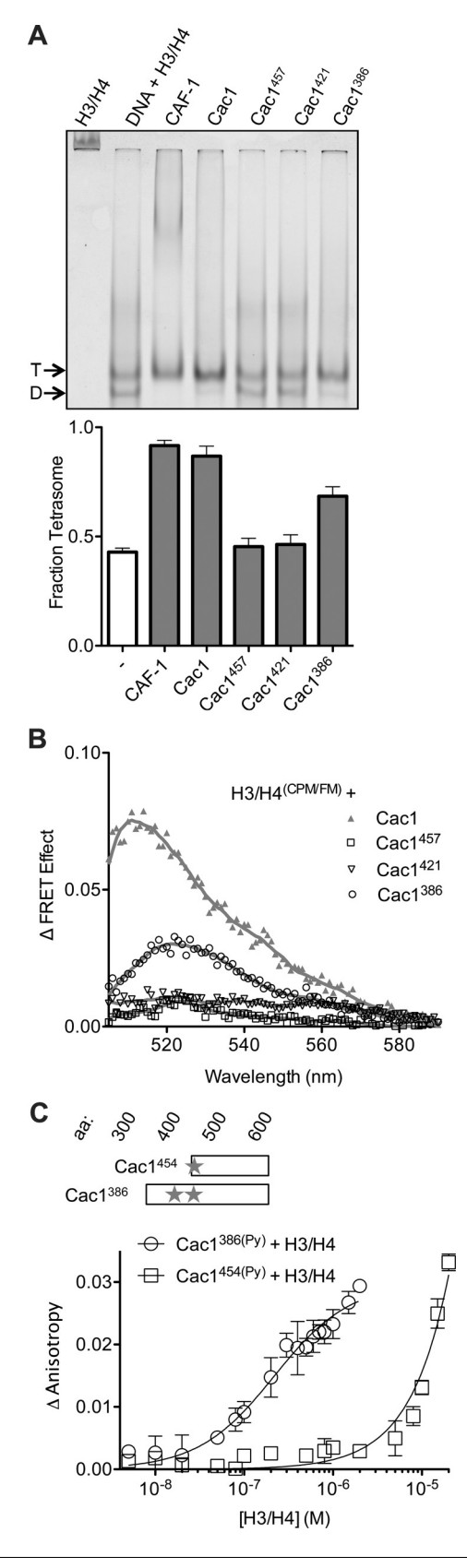

**Figure 4.** The C-terminus of Cac1 binds and
*Figure 4 continued on next page*

interactions between Cac1 and H3/H4. To test this, we used fluorescence anisotropy with pyrene-labeled Cac1[386] and Cac1[454] (Cac1[386(Py)] and Cac1[454(Py)], respectively) to monitor pyrene anisotropy changes with H3/H4 binding (*Figure 4C*). In this assay, Cac1[386(Py)] binds H3/H4 ($K_{Dapp}$ = 210 nM; *Table 1*) with a slightly weaker affinity than full-length Cac1 binding to H3/H4[(Py)] ($K_{Dapp}$ = 97 nM). Cac1[454(Py)] binding to H3/H4, on the other hand, was significantly weaker (*Table 1*). Together, these data reveal that Cac1[386] binds H3/H4 and is sufficient to promote histone tetramerization.

## The Cac1 C-terminus exhibits binding and conformational plasticity

Although intact CAF-1 does not dimerize during the H3/H4 tetramerization process (*Liu et al., 2012*; *Winkler et al., 2012b*), previous work found that the *H. sapiens* and *X. laevis* homologs of the Cac1 subunit alone can dimerize in vivo through a sequence located C-terminal to the conserved ED domain (*Quivy et al., 2001*; *Gérard et al., 2006*). Although this putative dimerization sequence is apparently not conserved in yeast, purified *S. cerevisiae* Cac1 elutes from size exclusion chromatography at a molecular weight consistent with a dimer, confirmed by multiangle light scattering (*Winkler et al., 2012b*).

To gain structural insight into the putative Cac1 dimer, we crystallized *E. coli*-expressed Cac1[457], a region that forms many stable intra-Cac1 chemical cross-links (*Figure 3—figure supplement 1*). This region also remains as a stable proteolytic fragment after expression and purification of Cac1 from baculovirus-infected Sf9 cells (*Figure 5—figure supplement 1*). The structure was determined using molecular replacement to a resolution of 2.9 Å (*Table 4*) with coordinates from a recently determined structure of Cac1C (522–600) (*Zhang et al., 2016*). Even though the entire 457–606 protein was present in the crystal (*Figure 5A* and data not shown), only amino acids 520–600 were visible in our electron density, and the 'wing' (576–581) was disordered. As expected, the structure we determined is virtually identical to the reported Cac1C winged helix (WH) domain, with an overall root mean squared deviation (r.m.s.d.) of 1.14 Å for all atoms (*Zhang et al., 2016*). Indeed, we observed a crystallographic homo-dimer. Amino acids 572–600 – corresponding to the last beta strand, loop, and alpha helix – form a symmetrical head-to-tail homo-dimerization interface (*Figure 5A—figure*

*Figure 4 continued*

tetramerizes H3/H4. (**A**) EMSA evaluating tetrasome formation by Cac1 N-terminal truncations Cac1$^{386}$, Cac1$^{421}$ and Cac1$^{457}$ in H.B. The graph shows the mean and standard deviation from at least three independent experiments. Arrows point to complexes of DNA bound to H3/H4 dimers (D) or tetramers (T), respectively. (**B**) Change in FRET Effect of H3/H4$^{(CPM/FM)}$ induced by 2 µM Cac1$^{386}$, Cac1$^{421}$ or Cac1$^{457}$. The Cac1 spectrum is included from *Figure 2B* for reference. (**C**) Fluorescence anisotropy of Cac1$^{386(Py)}$ or Cac1$^{454(Py)}$ titrated with H3/H4 in H.B. The schematic indicates two labeled residues on Cac1$^{386}$ (cysteines 440 and 454), and one on Cac1$^{454}$.

The following figure supplement is available for figure 4:

**Figure supplement 1.** Histone deposition assay of Cac1 truncations in Minimal Buffer (M.B.).

*supplement 2* and *Supplementary file 1D*) that encompasses 651 Å$^2$ of solvent-inaccessible surface area.

In order to examine dimerization of the Cac1 C-terminus in solution, we monitored the fluorescence anisotropy of Cac1$^{386(Py)}$ or Cac1$^{454(Py)}$ with titration of the same unlabeled Cac1. In a minimal buffer (M.B.), titration of unlabeled Cac1$^{386}$ into Cac1$^{386(Py)}$ increased anisotropy with a dissociation constant of 26 nM (*Figure 5B* and *Table 1*). Likewise, unlabeled Cac1$^{454}$ titrated into Cac1$^{454(Py)}$ increased pyrene anisotropy with a similar equilibrium constant ($K_D$ = 25 nM). The results are consistent with Cac1 forming a dimer through the C-terminus. However, we also found that the Cac1$^{386(Py)}$ - Cac1$^{386}$ and Cac1$^{454(Py)}$ - Cac1$^{454}$ dimers are destabilized by the histone buffer (H.B.) that is typically used in H3/H4 equilibrium binding experiments (*Donham et al., 2011*; *Scorgie et al., 2012*; *Karantza et al., 1996*; *Banks and Gloss, 2004*; *Winkler et al., 2012a*; *Winkler et al., 2012b*).

Importantly, our histone tetramerization assays performed in H.B. (*Figure 4A and B*, *Figure 4—figure supplement 1*) indicate that monomeric Cac1$^{386}$ is sufficient to tetramerize (H3/H4)$_2$ but Cac1$^{454}$ is not, even though both harbor the putative dimerization domain. Therefore, Cac1 dimerization is not necessary to promote (H3/H4)$_2$ tetramerization, consistent with the functional stoichiometry of CAF-1.

In addition to H3/H4 binding, the Cac1 C-terminus has been reported to bind to DNA with a dissociation constant of ~2 µM (*Zhang et al., 2016*) and found here to possess many cross-links to Cac2 (*Figure 3A and B*). We also found that Cac1$^{386(Py)}$ directly binds Cac2 with a $K_D$ of 1.3 µM in H.B. (*Figure 5C* and *Table 1*). The weak binding of the Cac1 C-terminus to multiple partners, along with the histone-induced HX changes and simultaneous cross-links to both H3/H4 and Cac2, suggests that this region has binding and conformational plasticity.

The HX, CX, and anisotropy measurements of Cac1 with H3/H4 (*Figures 2A*, *3B* and *4C*) indicate that structural changes take place C-terminal to the Cac1 ED domain. To examine this possibility, we used pyrene as a photophysical probe for conformational changes. Cac1$^{386(Py)}$ harbors two pyrene-labeled cysteines (residues 440 and 454) in this region (*Figure 4C*), whereas Cac1$^{454(Py)}$ only has one. Two pyrenes that are within 10 Å exhibit a characteristic 'excimer' band in the 465 nm region of the fluorescence spectrum (*Birks et al., 1963*). This band was observed for monomeric Cac1$^{386(Py)}$ in H.B. but not for Cac1$^{454(Py)}$, as expected (*Figure 5D*). To detect conformational changes in the vicinity of these cysteines, we monitored the Cac1$^{386(Py)}$ excimer band with a saturating concentration of either H3/H4 or Cac2. We found that binding of H3/H4 removes this excimer band, whereas Cac2 does not change the spectrum. These results indicate that direct binding of H3/H4, but not Cac2, promotes a structural change in Cac1 near the ED domain.

## Discussion

The H3/H4 chaperone CAF-1 is conserved among all eukaryotes and has important roles during DNA replication and repair, regulation of gene expression, and maintenance of chromatin accessibility. The broad range of approaches we present here reveal architectural and mechanistic insights into CAF-1 and CAF-1-H3/H4 complexes that are relevant for histone chaperone structure and function. These results shape a new, detailed model of the CAF-1-H3/H4 complex (*Figure 6*) that exhibits both modular and cooperative interactions involved in the assembly of (H3/H4)$_2$ tetramers during DNA replication.

**Table 4.** Data collection and refinement statistics.

| | |
|---|---|
| Wavelength | 1.0 Å |
| Resolution range – data collection | 29.43–2.91 (3.01–2.91) |
| Space group | P 41 2 2 |
| Unit cell (Å) (deg) | 58.850 58.830 97.929 90 90 90 |
| Total reflections | 26,419 (5001) |
| Unique reflections | 4117 (393) |
| Multiplicity | 6.42 (6.63) |
| Completeness (%) | 99.3 (99.7) |
| Mean I/sigma(I) | 12.9 (1.7) |
| Wilson B-factor | 92.56 |
| R-meas | 0.099 (0.557) |
| Resolution range - refinement | 29.43–3.00 (3.107–3.00) |
| Reflections used in refinement | 3761 (365) |
| Reflections used for R-free | 360 (42) |
| R-work | 0.233 (0.408) |
| R-free | 0.275 (0.324) |
| Number of non-hydrogen atoms | 654 |
| Macromolecules | 653 |
| Protein residues | 81 |
| RMS(bonds) | 0.007 Å |
| RMS(angles) | 0.93 deg |
| Ramachandran favored (%) | 88 |
| Ramachandran allowed (%) | 12 |
| Ramachandran outliers (%) | 0 |
| Rotamer outliers (%) | 4.3 |
| Clashscore | 6.85 |
| Average B-factor | 48.8 |
| Number of TLS groups | 3 |

Statistics for the highest-resolution shell are shown in parentheses.

Friedel mates were averaged when calculating data collection statistics.

## Dynamic and modular architecture of CAF-1

Our investigation into CAF-1 inter-subunit interactions has revealed a central role for the large Cac1 subunit in CAF-1 function. Previous studies show that both yeast and metazoan homologs of the large subunit directly interact with the other subunits, but no direct interaction between the mid-sized and small subunits has been reported (*Tyler et al., 2001*; *Kaufman et al., 1995*). These observations are born out in the chemical cross-linking experiments (*Figure 3A and B*), which show extensive cross-links between Cac1-Cac2 and Cac1-Cac3, but not between Cac2-Cac3.

The Cac1 residues (lysines 235 and 282) that cross-link to Cac3 flank a Cac1 L276P mutation that abolished Cac3 binding, and are also located within a region (residues 215–429) known to associate with Cac3 in vitro and in vivo (*Krawitz et al., 2002*). Just N-terminal to the Cac1-Cac3 cross-links, Cac1 harbors the PCNA interacting peptide box (PIP-box; residues 225–232) (*Rolef Ben-Shahar et al., 2009*). Notably, extensive intra-Cac1 cross-links surround these functional sites. Therefore, we refer to residues 118–334 as the 'middle domain' of Cac1 (*Figure 6*, and *Supplementary file 1C*). Our architectural model places the middle domain at the center of all protein-protein interactions within the CAF-1 and CAF-1-H3/H4 complexes, leading to the idea that it coordinates many

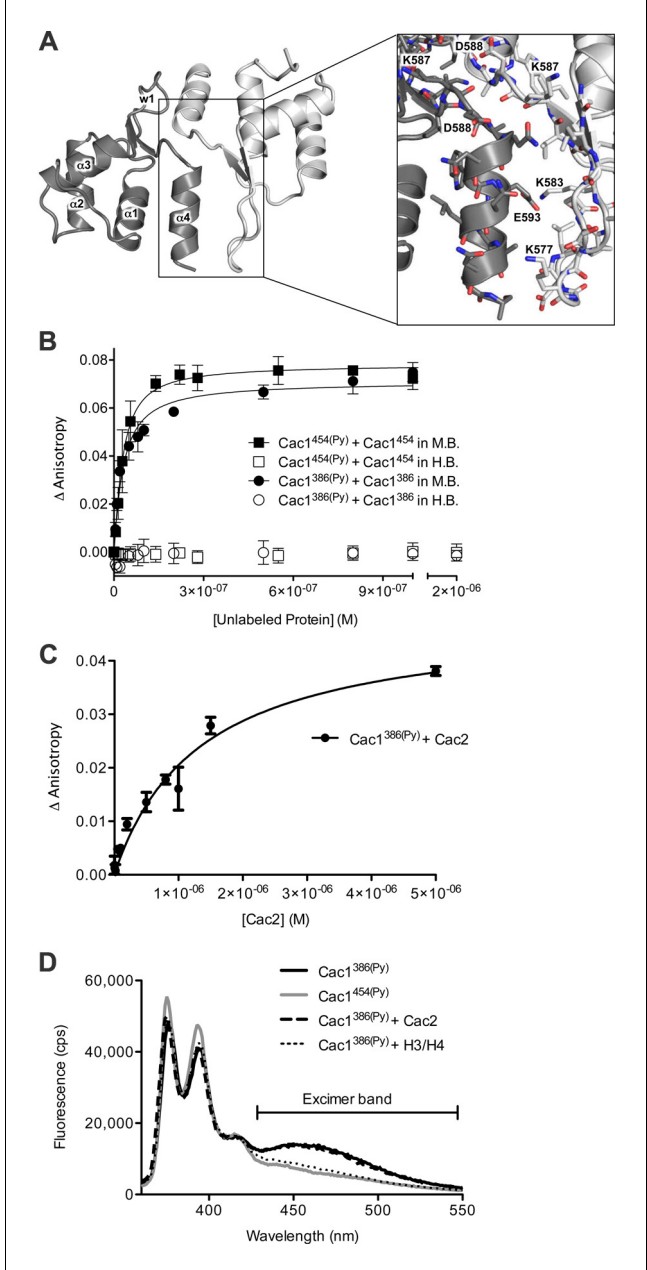

**Figure 5.** The Cac1 C-terminal winged helix (WH) domain can form a homodimer. (**A**) Crystal structure of amino acids 520–600 at a resolution of 2.9 Å (PDB ID 5JBM), shown as two crystallographically related monomers colored separately (light gray and dark gray). The inset shows major interacting residues buried in half of the homodimer interface, which is arranged in a head-to-tail symmetry with identical interactions on both halves. (**B**) Homo-dimerization of the Cac1 C-terminus quantified by titrating unlabeled Cac1$^{386}$ or Cac1$^{454}$ into 10 nM of labeled Cac1$^{386(Py)}$ or Cac1$^{454(Py)}$, respectively. The pyrene anisotropy of Cac1$^{386(Py)}$ or Cac1$^{454(Py)}$ increases in Minimal Buffer (M.B.: 20 mM HEPES, 150 mM NaCl, 1 mM DTT, pH 7.5), but homo-dimerization does not occur in H.B. (**C**) Binding affinity of the Cac1$^{386(Py)}$-Cac2 interaction. Pyrene fluorescence anisotropy of 10 nM Cac1$^{386(Py)}$ titrated with increasing concentration of Cac2 in Histone Buffer (H.B.). The $K_D$ was determined to be 1.3 µM (**Table 1**). (**D**) Pyrene fluorescence spectra of Cac1$^{386(Py)}$ alone, Cac1$^{454(Py)}$ alone, and Cac1$^{386(Py)}$ bound to 2 µM H3/H4 or 13 µM Cac2. The excimer band that peaks at 465 nm is indicated.

The following figure supplements are available for figure 5:

**Figure supplement 1.** Purification of full-length Cac1 and Cac1$^{457}$ from baculovirus-infected Sf9 cells.

*Figure 5 continued on next page*

*Figure 5 continued*

**Figure supplement 2.** Structural analysis of the Cac1[457] WH domain.

functions during replication, such as recruitment to the fork through PCNA and stabilizing CAF-1 architecture.

The cross-links detected between Cac1 and Cac2 indicate a more extensive binding mode than previously seen. The human Cac1 and Cac2 homologs – p150 and p60, respectively – require the C-terminal one-third of p150 for binding (*Kaufman et al., 1995*). In addition to Cac2 cross-links at the Cac1 C-terminus, we observed several Cac2 cross-links to the Cac1 N-terminus and middle domain (*Figure 3A and B*). As Cac1[386(Py)] binds Cac2 very weakly (*Table 1* and *Figure 5C*), both the N-terminus and middle domain also likely contribute to Cac2 binding.

Interestingly, we observed different Cac1-Cac2 cross-links in the presence of H3/H4, with most occurring in Cac1 regions that also cross-link to H3/H4. Cac2 undergoes many changes in HX with H3/H4 binding to CAF-1 (*Figure 3A and B*), some of which coincide with sites of Cac1 cross-links. For example, the C-terminus of Cac2 cross-links just N-terminal to the WH domain, and becomes more accessible to HX with H3/H4 binding (residues 427–443, *Figures 3A* and *2A*). In contrast, Cac2 residues 284 and 285 cross-link to the WH domain but are among the most protected from HX with

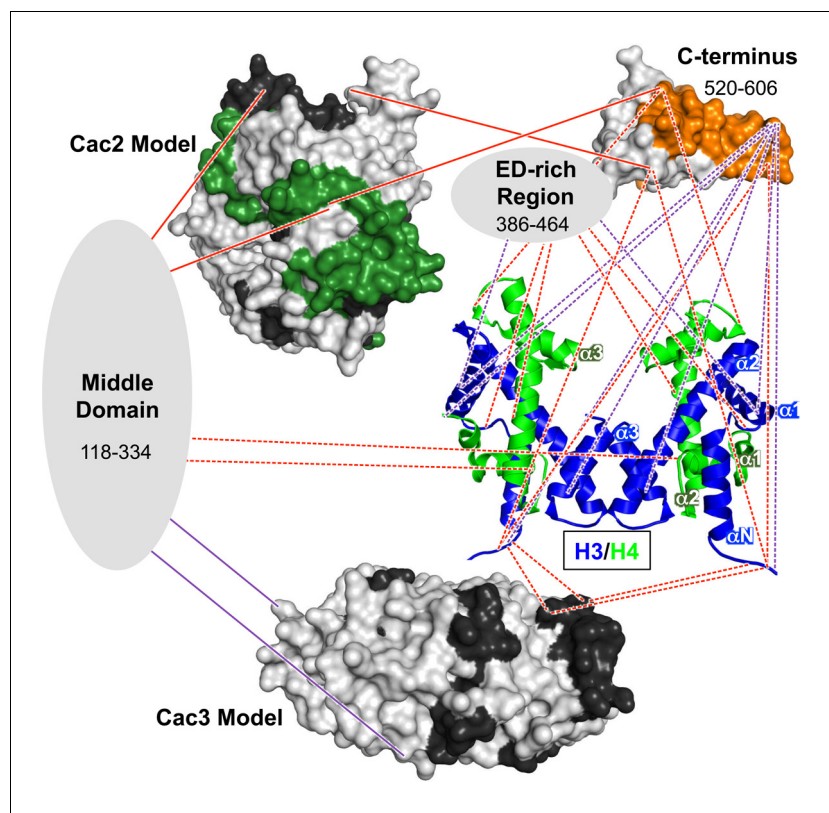

**Figure 6.** Architectural model of the CAF-1-H3/H4 complex. Cac2 and Cac3 are presented using PHYRE2 models. Cac1 is presented with respect to the domains observed in this study: the 'Middle Domain,' which consists of amino acids 118–334 and includes the KER region; the 'ED-rich Region' that includes the ED domain and the adjacent amino acids; and the 'C-terminus' that includes the WH domain. The nucleosomal (H3/H4)$_2$ tetramer is shown (1ID3.pdb) with H3 (blue) and H4 (bright green) colored to distinguish the histones. CAF-1 proteins are colored according to the 60' HX data and coloring scheme in *Figure 2B*. The cross-linking data is incorporated using the same coloring and line schemes as in *Figure 3B*.

H3/H4 (*Figures 2A*, *3A and B*). This suggests that conformational changes in Cac2 take place upon histone binding.

C-terminal deletions of human p150 have detrimental effects on p150-mediated chromatin assembly and binding to p60 (*Kaufman et al., 1995*). Our data support the importance of the Cac1 C-terminus for these functions (*Figures 4A,B* and *5C*), as well as in the CAF-1-dependent response to DNA double-strand breaks (*Figure 2C*). The C-terminus exhibits binding plasticity for context-dependent interactions with Cac2, H3/H4, and DNA (*Zhang et al., 2016*) (*Figures 4C* and *5C*). The cross-links from the H3/H4 N-terminal tails to the WH domain (E569, K577, D579, K583, D588, and E593) (*Figure 3B*; *Supplementary file 1C*) coincide with Cac1 residues that have been reported to interact with DNA. Mutations to residues in the same positively-charged surface (K560E, K564E, K568E, R573E and R582E) had detrimental effects on WH-DNA binding and functional consequences in vivo (*Zhang et al., 2016*). Thus, the WH domain exhibits binding plasticity for different partners.

These dynamic changes in Cac1 conformation are supported by distance-dependent changes in the pyrene spectrum of labeled cysteines 440 and 454 when bound to H3/H4 (*Figure 5D*), as well as H3/H4-dependent HX changes at Cac1 residues 550–591 (*Figure 2A*). The increase in HX upon H3/H4 binding is unexpected for a Cac1 region that also cross-links to histones. The weak affinity of Cac1$^{454(Py)}$ to H3/H4 ($K_{Dapp}$ = n.c.; *Table 1*), however, suggests that the HX changes may not be due to direct binding, but rather to allosteric conformational changes or conformational mobility dependent on other binding site(s) on Cac1 (*Engen, 2009*). Together, these results shape a functional model, wherein Cac1 serves as a central scaffold with a modular architecture, which is subject to conformational changes upon H3/H4 binding.

## Interactions of CAF-1 with H3/H4

Of the three CAF-1 subunits, Cac1 is uniquely capable of both binding to H3/H4 and tetramerizing histones (*Figure 1B–D*). CAF-1 primarily interacts with H3/H4 through the last third of Cac1 (*Figures 3B* and *6*), with the majority of these Cac1-H3/H4 cross-links occurring in the ED domain and the C-terminus to the cores and N-terminal tails of both H3 and H4. However, the Cac1-H3/H4 binding constant is more than an order of magnitude weaker than that of intact CAF-1 (*Winkler et al., 2012b*; *Liu et al., 2012*) (*Table 1*). Therefore, the other subunits contribute to H3/H4 binding through direct interactions as seen in the cross-linking study (*Figure 3B*), and also possibly indirectly through interactions that stabilize Cac1.

Several functional consequences for H3/H4 mutants linked to CAF-1 have been observed in yeast, including mutants that derepress rDNA, telomeric, and mating loci silencing (reviewed in [*Li and Zhang, 2012*]). CAF-1 is known to co-purify with specific post-translational modifications (PTMs), namely H4 acetylated lysines 5, 8, 12, 16, and H3 acetylated lysine 56 and methylated (me$^1$, me$^2$, and me$^3$) lysine 79 (*Zhou et al., 2006*; *Masumoto et al., 2005*). Mutations to the acetylatable lysines in H4 have a detrimental effect on CAF-1-dependent H3/H4 incorporation into chromatin (*Glowczewski et al., 2004*). Moreover, a H3 K14R mutant, which cannot be acetylated, decreases levels of bound Cac2 and has functional consequences in rDNA silencing and aging (*Xu et al., 2016*). However, deletion of the H3 and H4 N-terminal tails did not alter CAF-1-H3/H4 binding affinity (*Winkler et al., 2012b*), nor histone deposition by CAF-1 (*Shibahara et al., 2000*). The cross-links from the H3/H4 tails to the WH domain, together with the weak Cac1$^{454(Py)}$ – H3/H4 affinity (*Figures 3B*, *4C* and *Table 1*), are consistent with these prior observations. As the WH domain interacts with histone tails, our model reveals how histone N-terminal PTMs could modulate CAF-1 function in vivo.

In contrast to the tails, the core domains of both histone H3 and H4 cross-link to multiple regions in Cac1, including the middle domain, WH domain, and regions near the ED domain. The cross-links in the vicinity of H3 K56 provide an explanation for the modulatory effect of H3 K56 acetylation on CAF-1 binding (*Li et al., 2008*; *Winkler et al., 2012b*).

The direct cross-links of Cac1-Cac3 occur in a region near where Cac3 homologs from multi-cellular organisms bind helix α1 of H4 (*Nowak et al., 2011*; *Murzina et al., 2008*). However, interactions previously observed between the H3 N-terminus with the top surface of the beta propeller (*Schmitges et al., 2011*) do not coincide with any cross-links observed here. The *D. melanogaster* homolog of Cac3 binds to the N-terminal peptides of either H3 or H4 with tight to modest affinity, respectively ($K_D$ = 35 nM for H4, $K_D$ = 2 μM for H3) (*Nowak et al., 2011*). That both of these interactions are tighter than our $K_D$ for Cac3 binding to H3/H4$^{(Py)}$ (*Figure 1B*) raises the possibility that

yeast Cac3 might bind H3/H4 differently than other eukaryotic homologs. The weak Cac2 and Cac3 interactions with histones measured here, together with the inability to tetramerize H3/H4, is consistent with the dearth of specific Cac2-H3/H4 and Cac3-H3/H4 cross-links.

Importantly, our cross-links are largely consistent with a recent report examining CAF-1 and CAF-1-H3/H4 cross-links (*Kim et al., 2016*). In both studies, the Cac1 middle domain cross-links to Cac2 and Cac3, and the Cac1 C-terminus cross-links to Cac2 and H3. We report an additional cross-link between the middle domain to H4 whereas Kim, et al. observe the middle domain cross-linking to H3, consistent with the conclusion that this domain scaffolds binding to Cac2, Cac3, and H3/H4. The electron microscopy data also support the idea that Cac1 is a platform for these interactions, as Cac1 is physically associated with Cac2, Cac3, and H3/H4 (*Kim et al., 2016*).

We observe more histone cross-links to the Cac1 C-terminus whereas Kim, et al. observe more to the middle domain. These differences may be attributed to experimental and computational differences. Specifically, cross-links are sensitive to cross-linking reaction times and amounts or concentrations of proteins and cross-linking agents. It is not possible to compare the differences between the relevant conditions in the two studies, as these methods are not well-detailed in the Kim et al. study (*Kim et al., 2016*; *Leitner et al., 2014*). Importantly, we emphasize that chemical cross-linking is a capture method rather than an equilibrium one; thus, the abundance of cross-links is not suggestive of equilibrium binding constants. Because cross-links report on amino acid proximity and have little or no relation to the binding energy of protein-protein interactions, we employed biophysical approaches under equilibrium conditions to validate the hypothesis that the Cac1 C-terminus interacts with histones. Full-length Cac1 binds H3/H4 with a $K_{Dapp}$ of 97 nM (*Figure 1C*), similar to the $K_{Dapp}$ observed for Cac1$^{386}$-H3/H4 (210 nM; *Figure 4C*). The bulk of the Cac1-H3/H4 interactions that contribute to lower the free energy of binding, therefore, indeed occur at the C-terminus of Cac1.

We also used a different strategy for determination of cross-linked peptides. Kim, et al. used xQuest, which analyzes a mix of unlabeled and isotopic cross-linkers, and relies on false discovery rates for statistical confidence (*Leitner et al., 2014*). False discovery rates are more useful for larger sample sizes, while mixing in isotopic cross-linkers reduces the signal intensity and increases the search space. In contrast, we used the Batch-Tag Web function in Protein Prospector, which provides scores based on y and b ion matches for the overall cross-linked peptide complex, one cross-linked peptide only, and the other cross-linked peptide only. This analysis is useful for minimizing false positives in cross-linking experiments, as high-scoring peptides are often cross-linked to poorer-scoring peptides (*Trnka et al., 2014*). Knowing the score of the poorer-scoring peptide, along with confirming ion matches in the spectrum, are critical determinants for a high confidence assignment. This approach compares favorably to approaches that do not show the score of the poorer-scoring peptide (*Trnka et al., 2014*).

## H3/H4 tetramerization is mediated by the ED and WH domains of Cac1

Whereas the Cac1 WH domain can form a homodimer, this form is not required for (H3/H4)$_2$ tetramerization (*Figures 4A,B* and *5A*). The putative dimerization domain consists of residues 572–600 at the C-terminus, and does not appear to extend to amino acid 386, as Cac1$^{386(Py)}$ and Cac1$^{454(Py)}$ have similar dimerization constants (*Table 1* and *Figure 5B*). The dimer form, however, was not observed in H.B. Thus, the Cac1$^{386}$ and Cac1$^{454}$ monomer/dimer equilibrium can be significantly affected by changes in buffer composition, similar to the H3/H4 dimer/tetramer forms (*Donham et al., 2011*). This observation highlights the susceptibility of highly charged proteins, such as histones and their chaperone counterparts, to buffer modifications that impact protein oligomerization states. Since CAF-1 is a monomer in complex with H3/H4, the dimerization observed for Cac1 alone is likely masked by other subunit interactions in intact CAF-1.

Dissection of Cac1 function identified a specific region in the C-terminus that is required to tetramerize (H3/H4)$_2$ (*Figure 4A and B*). In the HX and CX analysis, lysines 444 and 464 cross-link to H3 and H4, respectively (*Figure 3C*), and residues 463–473 are protected from exchange with H3/H4 (*Figure 2A*). Importantly, the region required for tetramerization, between amino acids 386 and 421, overlaps the highly acidic ED domain (residues 383–436). Our studies reveal its role in CAF-1-induced (H3/H4)$_2$ tetramer formation, consistent with the observation that deletion of the p150 ED domain has chromatin assembly defects (*Kaufman et al., 1995*). Therefore, the Cac1 ED domain

interacts with H3/H4, which is similar to how other histone chaperones use acidic patches/surfaces for histone binding (*Das et al., 2010*).

## Conclusions

Collectively, the data support a model in which the CAF-1-H3/H4 architecture is organized by Cac1 through modular protein-protein interactions within the middle, ED, and C-terminal regions (*Figure 6*). The CAF-1 conformation may be significantly different with and without histones, as evidenced by the different inter-subunit cross-linking patterns between CAF-1 and CAF-1-H3/H4 samples (*Figure 3A and B*). In addition, extensive changes in HX occur throughout Cac1 with H3/H4 binding, even in regions not in direct contact with H3/H4 (*Figure 2A*). Together, these results suggest that H3/H4 binding induces large structural changes in CAF-1.

The routes that histones follow before reaching the nucleosome are guided by histone chaperones, which position the histones for proper interactions with enzymes, nucleosome remodelers, other histone chaperones, and DNA (*Liu and Churchill, 2012*). DNA replication is coupled to histone chaperone function, as passage of the replisome stalls or slows with depletion of the histone chaperones FACT, Asf1, or CAF-1 in vivo and in vitro (*Groth et al., 2007*; *Hoek and Stillman, 2003*; *Schlesinger and Formosa, 2000*). The conservative model of $(H3/H4)_2$ tetramer inheritance during replication (*Prior et al., 1980*; *Xu et al., 2010*; *Hoek and Stillman, 2003*), an important mechanism for epigenetic inheritance of histone marks, is explained by the tetramer form of histones maintained by CAF-1. In contrast to other known mechanisms, monomeric CAF-1 is capable of binding $(H3/H4)_2$ tetramers through a minimal bipartite region in the monomeric Cac1 subunit. The Cac1 subunit, then, is responsible for assembling $(H3/H4)_2$ tetramers, localizing CAF-1-H3/H4 to replication forks through PCNA, and scaffolding the architecture (*Shibahara and Stillman, 1999*; *Krawitz et al., 2002*) (*Figure 6*). These biological functions likely require many protein-protein interactions through different Cac1 domains. The interactions identified by this study indeed appear to affect CAF-1-H3/H4 structure and function, as mutations designed to disrupt them impair the CAF-1-dependent DNA damage response in vivo (*Figures 2C* and *3C*). Thus, our model presents a novel mechanism of H3/H4 binding by a histone chaperone, illuminating a unique tetramerization pathway experienced by histones during replication.

# Materials and methods

## Preparation of proteins

The expression and purification of *X. laevis* H3 and H4 with point substitutions H3 C110A and H4 T71C were carried out as before (*Scorgie et al., 2012*). The procedure for labeling residue 71C in histone H4 with CPM (Invitrogen), FM (Invitrogen), or N-(1-Pyrenyl) maleimide (Sigma) were as previously reported. Briefly, each fluorophore was individually incubated at 15x molar excess with H4 protein in denaturing buffer (20 mM HEPES, 6 M guanidine HCl, 0.5 mM TCEP, pH 7.25). Excess fluorophore was removed by centrifugation through Sephadex G-15 beads (Sigma). The remaining labeled H4 was then assembled with H3 by extensive dialysis into high salt buffer (10 mM Tris, 2 M NaCl, 1 mM EDTA, 0.5 mM TCEP, pH 7.5). Soluble protein was finally isolated through size exclusion chromatography.

Purification of *S. cerevisiae* CAF-1 from baculovirus-infected Sf9 cells was carried out as previously described (*Liu et al., 2012*). Briefly, Sf9 cells were co-infected for 48 hr with viral stocks for each CAF-1 subunit (Cac1 with a C-terminal Strep II epitope, Cac2 with a C-terminal $His_{6x}$ epitope, and Cac3 with a C-terminal FLAG epitope), each with an MOI of 1. The cell pellets were homogenized in 10 mM Tris pH 7.4, 350 mM NaCl, 1 mM DTT, 10 μg/mL DNase I, along with inhibitors for proteases (EDTA-free tablet; Roche) and phosphatases (1 mM $Na_3VO_4$ and 10 mM NaF). For Cac1 and Cac3 (both with C-terminal Strep II epitopes) purifications, the same procedure was followed. After Cac1 purification, MALDI MS was used to analyze the peaks pooled from SEC. The Cac2 purification from Rosetta 2 (DE3) pLysS cells (Novagen) was performed as previously described (*Liu et al., 2012*).

Cac1 N-terminal deletions to be inserted into the pGEX6P-1 plasmid possessed a S503E mutation, in order to reflect the phosphorylated S503 observed in our mass spectrometry experiments (data not shown). This mutation was introduced by site-directed mutagenesis (Quikchange II XL kit; Agilent Technologies) using the following primers:

Forward primer:

5' – C ATC GTCT CTA CCA TCC AAA AGA AGT AAT GAG GAC TTA CAG GCA CAG AC – 3'

Reverse primer:

5' – GT CTG TGC CTG TAA GTC CTC ATT ACT TCT TTT GGA TGG TAG AGA CGA TG – 3'

The N-terminal deletions were then generated with the following primers, which included forward primers with a BamHI restriction digestion site, as well as a reverse primer overlapping the C-terminal Strep II tag with an EcoRI digestion site:

Forward primers:

Residues 386–606: 5'- CGA GGA TCC TCT GAC GTT GAA TGG GTT AAT G – 3'

Residues 421–606: 5'- GTT GGA TCC GGA GAG TTT GAC GGG TTT CTA G – 3'

Residues 454–606: 5'- CGC GGA TCC TGC CTA AAA TCC AAT TTT GAA AAC – 3'

Residues 457–606: 5'- CGG GGA TCC TCC AAT TTT GAA AAC TTA TCA GAG GAA – 3'

Reverse primer:

5'- GGT GAA TTC CTA CTT TTC GAA CTG CGG GTG -3'

For crystallography, the following reverse primer was used to clone Cac1[457] without the C-terminal Strep II tag:

5'- ATG CGG CCG CTT ACA AAG ACG GGG TTG GCA TAT TTG -3'

The inserts were then ligated by T4 ligase into pGEX6P-1, which provides an N-terminal GST tag upstream of a PreScission protease cleavage site. After sequence verification, the plasmids were transformed into Rosetta 2 (DE3) pLysS cells for protein expression. 20 mL cultures were grown overnight prior to inoculation into 3 L of total culture. When the optical density at 600 nm reached 0.4, protein expression was induced with 1 mM IPTG and allowed to incubate for 3 hr at 32°C. The pellets were then harvested and flash frozen before protein purification. Subsequently, the pellets were resuspended in 20 mM Tris, 1 M NaCl, 2 mM DTT, pH 7.4 with 10 µg/mL DNase I and protease (EDTA-free tablet; Roche) and phosphatase (1 mM $Na_3VO_4$ and 10 mM NaF) inhibitors. The lysate was sonicated, and then clarified by centrifugation. The resulting supernatant was bound to glutathione Sepharose beads (Thermo Fisher) for 2 hr at 4°C before cleavage with PreScission protease overnight. The cleaved protein was bound to a StrepTactin Sepharose column (GE Healthcare) and washed extensively. This StrepTactin column step was skipped when purifying Cac1[457] for crystallography. The protein was eluted from StrepTactin beads with 10 mM Tris, 350 mM NaCl, 2 mM DTT, 2.5 mM d-Desthiobiotin, pH 7.4, and finally purified through a Superdex 75 column in 20 mM HEPES, 150 mM NaCl, 1 mM DTT, pH 7.4.

Cac1 truncations Cac1[386] and Cac1[454] were labeled with N-(1-Pyrenyl) maleimide by incubation with 50x molar excess fluorophore at 4°C overnight in 20 mM HEPES, 150 mM NaCl, 0.5 mM TCEP, pH 7.4. Excess dye was removed through G-15 Sephadex beads.

## Fluorescence spectroscopy

For all fluorescence spectroscopy assays, a FluoroLog-3 fluorometer (Horiba) with a thermostat set at 20°C was used. For fluorescence anisotropy measurements of H3/H4[(Py)], 750 µL of 25 nM H3/H4[(Py)] was equilibrated in Histone Buffer (H.B.: 20 mM Tris, 150 mM KCl, 2 mM $MgCl_2$, 1% glycerol, 0.5 mM TCEP, 0.05% BRIJ-35, pH 7.5). CAF-1 or the appropriate CAF-1 subunit was then titrated into H3/H4[(Py)]. For CAF-1 titrations, 5 nM of H3/H4[(Py)] was used. For fluorescence anisotropy measurements of Cac1[386(Py)] and Cac1[454(Py)], 750 µL of 10 nM labeled protein were incubated in the same buffer, then titrated with Cac1[386], Cac1[454], or H3/H4.

With the polarizers in place, pyrene was excited at 345 nm (slit width: 6 nm) and measured at 375 nm (slit width: 13 nm). The anisotropy was calculated according to *Equation 1*:

$$r = (I_{VV} - G^*I_{VH}) / (I_{VV} + 2G^*I_{VH}), \tag{1}$$

where G is the grating factor (G = $I_{HV}$ / $I_{HH}$).

Before and after titration of Cac2 or H3/H4 into Cac1[386(Py)], the pyrene fluorescence spectra were also collected with polarizers in place by exciting pyrene at 345 nm (slit width: 5 nm), and scanning the emission from 360–550 nm (slit width: 7 nm).

If the anisotropy experiment resulted in a significant degree of fluorophore quenching (>10% of pyrene fluorescence intensity) – as was the case for H3/H4[(Py)] binding to CAF-1, Cac1, Cac2, and

Cac3 – the following equation was used to obtain a corrected anisotropy value (*Dandliker et al.,* *1981*):

$$r = ((((A - A_f)/(A_b - A))^*(Q_f/Q_b)^*(A_b)) + A_f)/(1 + ((A - A_f)/(A_b - A)^*(Q_f/Q_b))), \quad (2)$$

where A is the observed anisotropy, $A_f$ is the anisotropy of free H3/H4$^{(Py)}$, $A_b$ is the anisotropy of saturated H3/H4$^{(Py)}$, $Q_f$ is the fluorescence intensity of free H3/H4$^{(Py)}$, and $Q_b$ is the fluorescence intensity of saturated H3/H4$^{(Py)}$.

The rationale for choice of binding equation was based on *Figure 7*, with several considerations. First, we made the assumption that the H3/H4 tetramerization equilibrium constant (D + D' to T) was negligible under the FA experimental conditions. This is a reasonable assumption because we have shown previously (*Donham et al., 2011*; *Liu et al., 2012*) that H3/H4 exists as dimers under our experimental buffer conditions. Here, we find no anisotropy change of H3/H4$^{(Py)}$ with addition of unlabeled H3/H4 up to a 10 µM concentration (data not shown). Thus, the tetramerization $K_D$ is very weak relative to the CAF-1-H3/H4 $K_D$. Second, we cannot measure K1 and K2 independently, and so report an apparent $K_{Dapp}$, which is best fit with a one-site ligand depletion model.

The apparent dissociation constants were calculated in GraphPad Prism (v. 5.0d) by *Equation 3* to account for ligand depletion:

$$Fi = 1 + (Fmax)^*(((K_D + [A^*] + [B]i) - sqrt(((K_D + [A^*] + [B]i)2) - (4^*[A^*]^*[B]i)))/2^*[A]))), \quad (3)$$

where i indicates the varying concentrations of unlabeled protein B that were titrated into the labeled protein A*.

For the mixed fluorophore FRET experiments, 750 µL of 10 nM of H3/H4 was prepared with half of the histone population labeled with the FRET donor CPM at H4 Cys$^{71}$, and the other half labeled with the FRET acceptor FM at H4 Cys$^{71}$, equilibrated in H.B. The Förster radius for the CPM-FM pair is 52 Å (*Wu and Brand, 1994*). To evaluate FRET, CPM was excited at 385 nm, and the emission spectrum recorded from 400–600 nm. In parallel, FM was excited at 491 nm, and the emission recorded from 500–600 nm. The FRET Effect was calculated from the "Enhanced fluorescence of acceptor" method described by Clegg (*Clegg, 1992*), which calculates the efficiency of FRET by observing the enhanced emission signal of the acceptor fluorophore upon donor excitation:

$$\text{FRET Effect} = F_{385}/F_{491} \quad (4)$$

Under our FRET system, $F_{385}$ is the extracted emission of FM when excited at 385 nm, while $F_{491}$ is the emission of FM when directly excited at 491 nm. The $F_{385}$ curve is extracted by first fitting the donor-only spectrum (H3/H4$^{(CPM)}$) excited at 385 nm to the dual-labeled spectrum (H3/H4$^{(CPM/FM)}$) excited at 385 nm. The fitted curve is then subtracted from the dual-labeled spectrum, producing the extracted acceptor spectrum. The FRET Effect is obtained when the extracted $F_{385}$ spectrum is normalized by the direct excitation of acceptor ($F_{491}$), which is especially valuable when an H3/H4 binding partner quenches the FM signal. All presented values derived from fluorescence data were obtained from at least three independent experiments.

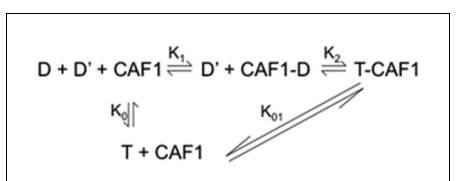

**Figure 7.** Potential equilibrium for CAF-1 association with H3/H4. D and T indicate dimers or tetramers of H3/H4, respectively.

## Electrophoretic mobility shift assay

0.2 µM H3/H4$^{(FM)}$ was incubated with 1.6 µM of the indicated histone chaperone or CAF-1 subunit in 10 mM Tris, 150 mM NaCl, 0.5 mM TCEP, pH 7.5 for at least 20 min on ice. For assays involving truncated Cac1, 2 µM of the Cac1 truncation was first incubated in either the above buffer or with H.B. for 30 min on ice prior to binding H3/H4$^{(FM)}$. After histone binding, 0.4 µM 80 bp Widom DNA was then introduced into the reaction for at least 20 min on ice. The disome/tetrasome species were separated by electrophoresis in 0.2x TBE (1x = 89 mM Tris,

89 mM boric acid, 2 mM EDTA, pH 8.0) 59:1 acrylamide:bis-acrylamide native gels for 150 min at 70 V. Fluorescein fluorescence was detected by scanning on a Typhoon 9400 imager (GE Healthcare) (excitation: 488 nm, emission: 526 nm). The Integrated Density Value (IDV) of the disome and tetrasome bands were quantified by ImageQuant (GE Healthcare), and the fraction of tetrasomes formed was calculated by *Equation 5*:

$$\text{Fraction Tetrasome} = \left(\text{IDV}_{\text{Tetrasomes}} - \text{IDV}_{\text{Tetrasome Background}}/2\right) / \left(\left(\text{IDV}_{\text{Disomes}} - \text{IDV}_{\text{Disome Background}}\right) + \left(\text{IDV}_{\text{Tetrasomes}} - \text{IDV}_{\text{Tetrasome Background}}/2\right)\right)$$

(5)

## Structural modeling

For homology modeling, the PHYRE2 (*Kelley et al., 2015*) server (Structural Bioinformatics Group, Imperial College) was used. The amino acid sequences of Cac2 and Cac3 without epitope tag sequence were input into the PHYRE2 web program (*Kelley et al., 2015*). 'Intensive' rather than 'Normal' modeling was used; 'Intensive' mode allows for use of multiple structural templates and *ab initio* modeling. Both Cac2 and Cac3 are confidently modeled as WD-repeat β propellers, with Cac2 possessing an additional disordered C-terminus (not shown) and Cac3 possessing an N-terminal helix.

## Hydrogen/deuterium exchange

CAF-1 and CAF-1-H3/H4 complexes were purified and buffer exchanged into 50 mM $KH_2PO_4$, 100 mM KCl, 5 mM DTT, pH 7.2. The samples were exchanged in $D_2O$ (99.9%, Cambridge Isotope Laboratories) at 10°C for varied lengths of time, then quenched with 100 mM dibasic potassium phosphate HCl, pH 2.4 at 0°C. The samples were then immediately injected into a 0°C chamber, digested using an online immobilized pepsin column and subjected to reversed-phase UPLC (Ultra high Performance Liquid Chromatography) for high resolution separation. To determine the extent of deuteration by mass, electrospray ionization and MSe data acquisition were implemented on a Synapt G2 (Waters Corp) q-TOF (Quadrupole-Time of Flight) mass spectrometer, which enabled high resolution and accurate mass analysis of both precursor and fragment ions.

A peptide list was generated by analyzing a mock experiment using undeuterated buffers resulting in unlabeled protein and searching the resulting data using the ProteinLynx Global Server (Waters Corp) search algorithm. DynamX v3.0 software (Waters Corp) allows automated detection of counterpart peptides identified in the unlabeled analyses within raw data files containing MS data for all deuterated peptides at each time point. DynamX was used to assign individual isotope distributions and compute weighted average mass values for deuterated peptides. All isotope assignments by DynamX were manually validated. No back exchange correction was used, because only relative changes in deuteration were calculated, which is expected to be identical for both raw and corrected deuteration levels (*Wales and Engen, 2006*).

## Yeast strains and plasmids

The genotypes of yeast strains and plasmids are described in *Table 2* and *Supplementary file 1B*. The sequences of primers used for mutagenesis are listed in *Supplementary file 1B*. Following confirmation by sequencing, the mutated plasmids were transformed into strains deleted for the endogenous *CAC* genes. The empty vector was the parent plasmid pRS315. Resistance to zeocin was determined by five fold dilution analysis of 1 OD 600 nm logarithmically growing cultures of yeast strains onto plates with and without the indicated amounts of zeocin. Following 2–3 days of growth at 30°C, the yeast plates were photographed.

## Chemical cross-linking

For DSS cross-linking, DSS (Thermo Pierce) was prepared to 10 mM in DMSO. 40 μg of the CAF-1 and CAF-1-H3/H4 complexes were each allowed to incubate with 500 μM DSS for 30 min at room temperature in 40 μL of 20 mM HEPES, 150 mM NaCl, pH 7.4. For EDC cross-linking, EDC was prepared to 75 mM in 50 mM $KH_2PO_4$, 100 mM KCl, pH 6.0. 40 μg of CAF-1 or CAF-1-H3/H4 were incubated with 25 mM EDC in the EDC buffer for 90 min at room temperature. As a negative control, an uncross-linked sample was included.

The cross-linking reactions were quenched by addition of 50 mM Tris, pH 7.4, for an additional 15 min. Subsequently, 2 μL of 1% ProteaseMAX surfactant (Promega) was added. The total volume

was brought up to 93.5 µL with 50 mM $NH_4HCO_3$. Cysteines were reduced with 1 µL of 0.5 M DTT at 55°C for 20 min, then alkylated with 2.7 µL of 0.55 M iodoacetamide for 15 min at room temperature in the dark. Prior to protease treatment, 1 µL ProteaseMAX surfactant was added to ensure a denaturing environment. Trypsin (Promega #5111) or trypsin/LysC (Promega #V507A) was prepared with 50 mM acetic acid to a final concentration of 1 µg/µL, while GluC (Promega #V1651) was prepared to 1 µg/µL with 50 mM $NH_4HCO_3$. Trypsin alone, trypsin/LysC, or trypsin and GluC were added to the reaction at a 1:20 protease:protein (w/w) ratio, and allowed to digest for 3 hr at 37°C. The reaction was then spun down and TFA (Trifluoroacetic Acid) was added to a final volume/volume concentration of 0.5%. The sample was desalted and concentrated with a C18-embedded ZipTip (Millipore). The peptides were then vacuum-concentrated using a SpeedVac (Thermo) and resuspended in 20 µL of 0.1% FA (Formic Acid). Samples were analyzed on the LTQ Orbitrap Velos mass spectrometer (Thermo Fisher Scientific) coupled with an Eksigent nanoLC-2D LC system. For sample injection, 8 µL of sample was loaded onto a trapping column (ZORBAX 300SB-C18, 5 × 0.3 mm, 5 µm) and washed with 2% ACN (acetonitrile), 0.1% FA at a flow rate of 10 µL/min for 10 min. The trapping column was then switched online with the nano-pump at a flow rate of 600 nL/min. Peptides were separated on an in-house made 100 µm i.d. × 150 mm fused silica capillary packed with Jupiter C18 Resin (Phenomex; Torrance, CA) over a 45 min gradient from 6% - 40% ACN. The flow rate was adjusted to 350 nL/min after 10 min to increase the effective separation of the peptides. Data acquisition was performed using Xcalibur (version 2.1) software. Higher energy collisional dissociation was used to produce the fragment ions in the linear ion trap from the precursor ions, which were measured in the Orbitrap mass analyzer. For every MS scan, the nineteen most intense ions were selected for fragmentation, and masses selected for fragmentation were then excluded for a duration of 120 s after a repeat count of 3. Orbitrap obtained raw files were converted to de-isotoped, centroided peak lists using an in-house script (PAVA, UCSF).

## Protein prospector determination of cross-links

Cross-linked peptides were identified through the Protein Prospector package (*Chu et al., 2010*). The uploaded spectra were searched for constant modifications including DSS and carbamidomethylated cysteine using the Batch-Tag Web function. Batch-Tag identifies small, unspecified peptide modifications, adapted in cross-linking scenarios to search for large modifications predicted to be cross-linked peptides. Variable modifications included oxidized methionine, and phosphorylated serine, threonine, and tyrosine. Tryptic peptides were included, allowing for two missed cleavages. The precursor mass tolerance was set to 12 ppm (parts per million), while fragment mass tolerance was 25 ppm. The Search Compare function was used to filter spectra; cross-linked peptide spectra were confirmed with a high score value (>20), low expect value (<1 × $10^{-5}$), and multiple b and y main sequence ion matches (*Supplementary file 1C*).

## Protein crystallography

Cac1[457] without the C-terminal Strep tag II was expressed and purified as described above. The protein was concentrated to 7.5 mg/mL and extensively dialyzed into 20 mM HEPES, 50 mM NaCl, 2 mM TCEP, pH 7.4. Protein crystals grew in 0.2 M sodium formate, 14% PEG 3350, 0.1 M Tris, pH 8.3 using the sitting drop vapor diffusion method. Data were collected at the Molecular Biology Consortium beamline 4.2.2 at the Advanced Light Source at Lawrence Berkeley National Laboratory, and processed using d*trek (*Pflugrath, 1999*) (*Table 4*). Molecular replacement was performed using PHASER (*McCoy et al., 2005*) implemented in Phenix (*Adams et al., 2010*) with PDB ID 5EJO (*Zhang et al., 2016*) as the search model. The placed model was refined through iterative refinement and model building using Phenix.refine and Coot (*Emsley and Cowtan, 2004*) until the Rfree value was sufficiently low for a model with good geometry, stereochemistry and structural characteristics as determined by PDB validation analyses (*Read et al., 2011*) (*Table 4*). The final structure has PDB ID 5JBM. The buried surface area and the contacts between Cac1[457] and the indicated symmetry related molecule were calculated using Areaimol and Contact programs implemented in the CCP4 suite of programs (*Bailey, 1994*), respectively. Superpose (*Maiti et al., 2004*) was used to measure R.m.s.d values. Electrostatics calculations were performed with CHARMM (*Jo et al., 2008*), and PyMol (*Schrödinger, LLC, 2014*) was used to generate structure figures.

## Acknowledgements

We thank the mass spectrometry and tissue culture cores at UC Denver for experimental advice and assistance, including Monika Dzieciatkowska, Chris Ebmeier, Lori Sherman, and Michelle Randolph. We are indebted to Sarita Namjoshi and Colleen Fisher for assistance with yeast plasmid mutagenesis and yeast strain construction. We are grateful to David Jones, Chris Malarkey, and Jean Scorgie for their technical support throughout this project, and Aaron Johnson for careful critique of this manuscript. We also appreciate the contributions of Dr. Jay Nix at beamline 4.2.2 at the Advanced Light Source, Lawrence Berkeley National Laboratory, as well as the Structural Biology and Protein Production, Monoclonal Antibody, Tissue Culture Shared Resources of the University of Colorado Cancer Center (NIH P30CA046934), NIH-NCATS Colorado Consortium CCTSA (UL1TR001082), and NIH Shared instrumentation grants (S10OD012033, S10OD12073, NIH S10 RR026641 and S10RR024599). This work was supported by funding from NIH R01GM11190 to MEAC.

## Additional information

### Competing interests

JKT: Senior editor, *eLife*. The other authors declare that no competing interests exist.

### Funding

| Funder | Grant reference number | Author |
|---|---|---|
| National Institutes of Health | P30CA046934 | Jessica K Tyler |
| National Institutes of Health | UL1TR001082 | Kirk C Hansen |
| National Institutes of Health | S10OD012033 | Mair EA Churchill |
| National Institutes of Health | S10OD12073 | Mair EA Churchill |
| National Institutes of Health | S10RR024599 | Kirk C Hansen |
| National Institutes of Health | R01GM11190 | Mair EA Churchill |
| National Institutes of Health | S10RR026641 | Wallace H Liu |

The funders had no role in study design, data collection and interpretation, or the decision to submit the work for publication.

### Author contributions

WHL, Conception and design, Acquisition of data, Analysis and interpretation of data, Drafting or revising the article; SCR, YZ, JCL, Acquisition of data; Z-JS, BKD, JLB, Acquisition of data, Analysis and interpretation of data; TN, Acquisition of data, Analysis and interpretation of data, Drafting or revising the article; NGA, Conception and design, Analysis and interpretation of data; KCH, Conception and design; JKT, Conception and design, Drafting or revising the article; MEAC, Conception and design, Analysis and interpretation of data, Drafting or revising the article

### Author ORCIDs

Jessica K Tyler, http://orcid.org/0000-0001-9765-1659
Mair EA Churchill, http://orcid.org/0000-0003-0862-235X

## Additional files

### Supplementary files

• Supplementary file 1. Supplementary tables. (A) Peptides identified in HX studies. (B) Primers used in the studies in yeast. (C) Chemically cross-linked peptides identified by XL-MS. (D) Cac1C putative dimer contacts.

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
