## [Decision Letter]

Thank you for submitting your article "The Cac1 Subunit of Histone Chaperone CAF-1 Organizes CAF-1-H3/H4 Architecture and Tetramerizes Histones" for consideration by *eLife*. Your article has been reviewed by two peer reviewers, and the evaluation has been overseen by a Reviewing Editor and Kevin Struhl as the Senior Editor. The following individuals involved in review of your submission have agreed to reveal their identity: Paul D. Kaufman (Reviewer #3).

The reviewers have discussed the reviews with one another and the Reviewing Editor has drafted this decision to help you prepare a revised submission.

Summary:

Overall, the referees are in agreement that the work represents an important advance in understanding the organization of the CAF-1 complex and how it mediates histone H3/H4 tetramerization. Providing the following issues can be suitably addressed, publication in *eLife* is recommended.

Essential revisions:

1) There needs to be better congruency between some of the biochemical results and the genetic tests. For example, in Figure 2, why are only a portion of the Cac1 regions identified as most altered in the HX experiments tested in the damage resistance experiments in Figure 2? Specifically, a 463-473 deletion was tested in vivo, not 254-276 or 340-360 deletions – why? Conversely, why was a 497-501 deletion made and tested? And near the C-terminus, why was a small 578-580 deletion made when the most altered HX region appears to be much larger (Figure 2—figure supplement 1)?

2) Since Cac1 addition causes tetramerization of H3/H4, the anisotropy of H3/H4 will be affected by histone assembly in addition to binding by Cac-1. How does this affect the Kd calculations, and what kind of model was used for determining apparent Kd's? In addition, several binding curves fail to reach saturation (e.g., Figure 4, as well as Cac2 and Cac3 of Figure 1). Kd calculations should not be attempted in such instances.

3) It is unclear whether the HX methodology typical of the Ahn lab was used to account for back-exchange and to establish how much deuterium uptake could actually be measured by each peptide examined (e.g., see Hoofnagle et al., 2001, PNAS). If not, what was the rationale for eliminating these critical steps? There would seem to be a problem here, because (for instance) all H3/H4 exchange should be complete by 30' based on results published by others, but the most that is seen for any histone peptide is about half exchange. And for many peptides it is much less. Please clarify.

4) Additional explanation would be helpful regarding the interaction between the C-terminus of Cac1 and H3/H4. Oddly, the HX increases here, rather than decreases (as one might predict as the simplest way to envision secondary structure stabilization upon binding). The crosslinking is compelling in suggesting that this is the interaction interface, so the surprising HX result could imply that some stabilizing Cac1 interaction in cis is broken when the protein binds to histones. Can the HXMS data help explain what is going on here? The authors may want to highlight how much exchange is going on at the two timepoints. Is HX now completely exchanged when Cac1 encounters histones, or is it completely protected without histones and is just starting to exchange when it binds to them? Even with the table in the supplement, not knowing how much of the measurement is due to back exchange versus not being exchanged in the first place (see point #3, above) makes it challenging to know what to conclude from these analyses.

5) Figure 2 is difficult to assess. The legend states that each bar is a peptide, but the figure instead seems to show a complicated series of interconnected shapes. Figure 2—figure supplement 1.is far easier to absorb (and would be even easier if the figure included a key to understand the coloring scheme). It is recommended that Figure 2 be replaced with a smaller version of Figure 2—figure supplement 1 (perhaps lacking some of the amino acid numbering in the peptide bars), but that the bigger version also be kept as a supplement.

6) The diagrams in Figure 2 should have the highlighted features (e.g. the N-terminus of Cac2 and blade numbers that are referred to in the main text) clearly labeled on the structural models. The color scheme here also needs to more clearly differentiate between areas where there is zero peptide coverage compared to areas where there no substantial difference in HX is measured at either/both of the two timepoints. Such a scheme could be helpful in rationalizing why there is no change at the H3/H3' 4-helix bundle; at first glance, one would think that this regions should gain protection going from a dimer to a tetramer (based on published HXMS results). It appears there are insufficient (any?) peptides that could be used to measure this histone interaction – can the authors comment why this might be?

7) What buffer is used for the histone alone sample for H3/H4 in the HX experiments? Was it a "histone buffer" that contains detergent? Does detergent affect the results here or with oligomerization? Does it account for the very sparse peptide coverage on the histones in the HX experiments?

8) Comparing Figure 2 and Figure 3: the rad52 strain appears surprisingly robust at 1.0 µg/ml Zeocin in Figure 2, but not 3C. Why might growth be so variable?

9) The authors should reference a recent work from Song and co-workers (Kim et al., Sci. Rep. 2016) that also performed chemical crosslinking of yeast CAF-1, and that also came to the conclusion that the Cac1 subunit is the platform for interactions with other subunits. In addition, the crosslinking of not only individual residues but whole domains appears to differ in many ways between the two papers. For one example, the present work detected Cac1-histone crosslinks on the C-terminal half of Cac1, but many histone crosslinks map to the N-terminal half of Cac1 in the Kim et al. paper. Are there differences in the protocols used consistent with these observations? Does this change some of the conclusions here, and why or why not? Also, how do the EM analyses in the Kim paper comport with the model in Figure 6?

10) Subsection “Dynamic and Modular Architecture of CAF-1”: "N-terminal deletions of human p150 have detrimental effects on p150-mediated chromatin assembly and binding to p60 (Kaufman et al., 1995)." Don't the authors mean C-terminal deletions have these detrimental effects?

11) Third paragraph, subsection “Interactions of CAF-1 with H3/H4”: This paragraph is somewhat confusing. Why are the WH-histone tail crosslinks surprising? Just because this region also contacts DNA, why does that exclude it from also (or alternatively, in different molecules in a population) interacting with histones?

---

## [Author Response]

*Essential revisions:*

*1) There needs to be better congruency between some of the biochemical results and the genetic tests. For example, in Figure 2, why are only a portion of the Cac1 regions identified as most altered in the HX experiments tested in the damage resistance experiments in Figure 2? Specifically, a 463-473 deletion was tested* in vivo*, not 254-276 or 340-360 deletions – why? Conversely, why was a 497-501 deletion made and tested? And near the C-terminus, why was a small 578-580 deletion made when the most altered HX region appears to be much larger (550-591, Sup. Figure 2)?*

We previously did not include all of the mutations that we made and tested, only those that expressed and had a phenotype. To address this point, we have now added a table (Table 3) detailing the Cac1, Cac2 and Cac3 mutants, including the rationale for their choice, their expression, and their phenotypes. In addition, we have clarified that the Cac1 and Cac2 mutants shown in Figure 2 relate to the HX results, whereas the Cac3 mutants in Figure 3 relate to the CX results, since Cac3 had virtually no changes in HX.

For consistency, we re-plated the same Cac1 mutants in the original manuscript, except for Cac1 Δ497-501. This deletion was originally designed because of a Cac1 K499 cross-link to Cac2, but we did not focus on this in the figure or the text, and the mutant does not relate to Figure 2. Thus, we agree that the Δ497-501 mutant was confusing, and it is now excluded from Figure 2. Cac1 Δ254-276 was not chosen, because this is the same region where Cac1 is already known to interact with PCNA and Cac3 (Rolef Ben-Shahar, et al. 2009, Krawitz, et al. 2002, respectively). In addition, the Δ280-284 mutant is in the same domain (Figure 3—figure supplement 1) and did not express Cac1. The Cac1 Δ578-580 mutant expressed and was sensitive to zeocin, but the larger deletion 575-606 did not express. Importantly, amino acids 578-580 are positioned in the loop of the WH domain. We stated in the manuscript that mutations, where possible, were designed in loops and not in putative structured regions, in order to retain structural integrity as much as possible. Given what we learned from the crystal structure, it is clear that both of these larger deletions would have disrupted the WH domain and most likely led to no/poor Cac1 expression.

The re-plating of Cac2 mutants now exclude the repetitive L316A/K318A mutant and Δ371-373, as the latter was not based on HX, but chosen because it is an intervening loop between Δ1-15 and L316E/K318A on the same surface patch of Cac2. To correlate better with the HX findings, we also include a C-terminal deletion (Δ425-468) because of HX changes at this region (Figure 2). A V273A/P275A/S276A/G277A mutant was also included; together with the S206A/A207G and L316E/K318E mutants, the HX changes around blades 5 and 6 (Figure 2) are now adequately represented. In summary, we believe that the entire figure is now much easier for readers to comprehend, as it presents only the HX results.

*2) Since Cac1 addition causes tetramerization of H3/H4, the anisotropy of H3/H4 will be affected by histone assembly in addition to binding by Cac-1. How does this affect the Kd calculations, and what kind of model was used for determining apparent Kd's?*

The reviewers bring up an important point that we did not emphasize. We do not know the actual binding model for this reaction, but a reasonable scheme is shown below (D represents an H3/H4 dimer, D’ represents a separate H3/H4 dimer, and T represents a (H3/H4)_2_ tetramer). We make the assumption that the T + CAF-1 reaction is negligible in our system, and this simplifies the scheme to individual dimers binding to CAF-1. This is a reasonable assumption based on several results. The histone tetramerization dissociation constant was previously reported to be ~9 µM (Baxevanis, et al. 1991; reviewed in Rippe, et al., 2008) in low ionic strength and neutral pH. We have shown previously (Donham et al., 2011; Liu et al., 2012) that H3/H4 exist as dimers under our experimental buffer conditions. Here, we find no fluorescence anisotropy (FA) change of H3/H4^(Py)^ with titration of unlabeled H3/H4 up to 10 µM concentration (data not shown). These results suggest that the tetramerization K_D_ is very weak relative to the CAF-1-H3/H4 K_D_.

Thus, if only K_1_ and K_2_ are under consideration, then the following can be described: if K_1_ and K_2_ are independent and different, then we expect a two step binding isotherm; if K_1_ and K_2_ are independent and similar, we expect the appearance of a single site binding isotherm; if K_1_<K_2_, we expect positive cooperativity; if K_1_>K_2_, we expect negative cooperativity. We are operating in the regime where ligand depletion must be considered in the calculations, and we did not observe any isotherms that deviated significantly from the single-site binding model under conditions of ligand depletion (Equation 3). Perhaps this is due to the inability to observe the FA change for the “second dimer.” The FA change for D (27 kDa) to the D-CAF-1 complex (198 kDa) is relatively large, compared to the difference between D-CAF-1 to DD’-CAF-1 (225 kDa), which might render the second binding event unobservable. As such, we have added to the Methods section the rationale for the choice of model (in quotes below). In addition, to indicate that the CAF-1-H3/H4, Cac1-H3/H4, and Cac1^386^-H3/H4 binding constants represent apparent K_D_s, we now denote them as “K_D_app” in Table 1 and in the relevant text.

“The rationale for choice of binding equation was based on Scheme 1, with several considerations. First, we made the assumption that the H3/H4 tetramerization equilibrium constant (D + D’ to T) was negligible under the FA experimental conditions. This is a reasonable assumption because we have shown previously (Donham et al., 2011; Liu et al., 2012) that H3/H4 exists as dimers under our experimental buffer conditions. Here, we find no anisotropy change of H3/H4^(Py)^ with addition of unlabeled H3/H4 up to a 10 µM concentration (data not shown). Thus, the tetramerization K_D_ is very weak relative to the CAF-1-H3/H4 K_D_. Second, we cannot measure K1 and K2 independently, and so report an apparent K_Dapp_, which is best fit with a one-site ligand depletion model.”

*In addition, several binding curves fail to reach saturation (e.g., Figure 4, as well as Cac2 and Cac3 of Figure 1). Kd calculations should not be attempted in such instances.*

We agree with this suggestion and have denoted the dissociation constants of the following interactions as not calculated (“n.c.”): H3/H4^(Py)^ + Cac2, H3/H4^(Py)^ + Cac3, Cac1^454(Py)^ + H3/H4.

*3) It is unclear whether the HX methodology typical of the Ahn lab was used to account for back-exchange and to establish how much deuterium uptake could actually be measured by each peptide examined (e.g., see Hoofnagle et al., 2001, PNAS). If not, what was the rationale for eliminating these critical steps? There would seem to be a problem here, because (for instance) all H3/H4 exchange should be complete by 30' based on results published by others, but the most that is seen for any histone peptide is about half exchange. And for many peptides it is much less. Please clarify.*

No back exchange correction was used in our experiments, which is why expected exchange was lower than expected. Instead, raw deuteration was used. Given that we are examining relative changes only, the results from using raw deuteration and corrected deuteration are expected to be the same. This consideration is reviewed elsewhere (Wales and Engen, 2006). Our approach makes the assumption that back exchange is identical for a given peptide in the complexed and uncomplexed forms, which is reasonable considering that relative changes are measured in many examples of ligand binding.

We now mention this clearly in our Methods section.

“No back exchange correction was used, because only relative changes in deuteration were calculated, which is expected to be identical for both raw and corrected deuteration levels (Wales and Engen, 2006).”

*4) Additional explanation would be helpful regarding the interaction between the C-terminus of Cac1 and H3/H4. Oddly, the HX increases here, rather than decreases (as one might predict as the simplest way to envision secondary structure stabilization upon binding). The crosslinking is compelling in suggesting that this is the interaction interface, so the surprising HX result could imply that some stabilizing Cac1 interaction in cis is broken when the protein binds to histones. Can the HXMS data help explain what is going on here?*

We agree that the increase in HX at the Cac1 C-terminus with H3/H4, along with cross-links to H3/H4, might confuse readers. Our equilibrium binding experiments, however, point to the idea that Cac1 residues 454-606 can bind to Cac2 and H3/H4, in addition to the DNA interactions reported previously (Zhang, et al. 2016). These interactions are generally in the low µM range or weaker, suggestive of binding and conformational plasticity in this part of the protein. The cross-links are derived from a non-equilibrium method that has little relationship to binding energy. Thus, in the context of a CAF-1 complex, Cac1 residues 454-606 may contribute weakly to the binding energy with H3/H4, whereas residues 386-453 contribute more (Cac1^386^-H3/H4 K_Dapp_ = 210 nM; Cac1^454^-H3/H4 K_D_ = n.c.). With this in mind, the HX changes may not be due to direct histone binding, but could also be due to allosteric conformational changes or conformational dynamics leading to transient solvent penetration. This effect can also occur if solvent-exposed hydrogen-bonded secondary structures in the WH domain encounter a binding partner that alters hydrogen bonding (reviewed in Engen, 2009).

We have expanded and clarified this part of the Discussion, and have not quoted it here due to its length.

*The authors may want to highlight how much exchange is going on at the two timepoints. Is HX now completely exchanged when Cac1 encounters histones, or is it completely protected without histones and is just starting to exchange when it binds to them? Even with the table in the supplement, not knowing how much of the measurement is due to back exchange versus not being exchanged in the first place (see point #3, above) makes it challenging to know what to conclude from these analyses.*

This study reports differences in deuteration between samples at two timepoints and not the absolute levels of deuteration of the individual complexes as a function of time. There is little difference between the relative changes in deuteration observed for the 30 vs 60 minute time points. These experiments used preformed separate complexes of CAF-1 and CAF-1-H3/H4. They are each prepared and purified through SEC using the same buffer immediately prior to HX-MS. Therefore, we are not looking at the kinetics of the CAF-1-H3/H4 binding reaction. The issue of back exchange is addressed in point #3 above.

*5) Figure 2 is difficult to assess. The legend states that each bar is a peptide, but the figure instead seems to show a complicated series of interconnected shapes. Figure 2—figure supplement 1 is far easier to absorb (and would be even easier if the figure included a key to understand the coloring scheme). It is recommended that Figure 2 be replaced with a smaller version of Figure 2—figure supplement 1 (perhaps lacking some of the amino acid numbering in the peptide bars), but that the bigger version also be kept as a supplement.*

We agree that Figure 2 is hard to grasp given the overlapping bars. To improve the presentation, we have included a smaller version of Figure 2—figure supplement 1 to replace Figure 2, as suggested by the reviewers. In addition, we have re-analyzed the dataset – the numbers are slightly different and some peptides were removed, but there are no differences in the conclusions derived from the data. We have also removed the standard deviation column from [Supplementary-material SD1-data], because the experiments were run in duplicate.

*6) The diagrams in Figure 2 should have the highlighted features (e.g. the N-terminus of Cac2 and blade numbers that are referred to in the main text) clearly labeled on the structural models. The color scheme here also needs to more clearly differentiate between areas where there is zero peptide coverage compared to areas where there no substantial difference in HX is measured at either/both of the two timepoints.*

We thank the reviewers for this suggestion to clarify the figures dealing with HX. We have now labeled the relevant parts of the Cac2 structure model in Figure 2. In addition, in both Figure 2 and Figure 6, we now use a dark gray color to represent amino acids that lack HX coverage, in order to distinguish these residues from those that have coverage but did not exchange significantly.

*Such a scheme could be helpful in rationalizing why there is no change at the H3/H3' 4-helix bundle; at first glance, one would think that this regions should gain protection going from a dimer to a tetramer (based on published HXMS results). It appears there are insufficient (any?) peptides that could be used to measure this histone interaction – can the authors comment why this might be?*

Please see point 7 below.

*7) What buffer is used for the histone alone sample for H3/H4 in the HX experiments? Was it a "histone buffer" that contains detergent? Does detergent affect the results here or with oligomerization? Does it account for the very sparse peptide coverage on the histones in the HX experiments?*

We used a slightly different buffer with a different pH in order to stabilize the H3/H4 alone sample for the HX experiment. Therefore, the results between the H3/H4 and CAF-1-H3/H4 samples should not be compared, as the buffer differences alone may explain the differences in HX. Accordingly, we have removed the histone dataset from the manuscript. The HX data indeed had poor coverage, especially around the histone N-terminal tails and α3 of H3; however, the CX data has much better coverage and is sufficient to support our conclusion that CAF-1 interacts with both the N-terminal tails and α helical cores of H3 and H4. Removing these data do not significantly change any conclusions of this study.

*8) Comparing Figure 2 and Figure 3: the rad52 strain appears surprisingly robust at 1.0 µg/ml Zeocin in Figure 2, but not 3C. Why might growth be so variable?*

We agree that the Δrad52 yeast are more resistant to zeocin than expected in Figure 2. To ensure that our negative control is representative throughout this manuscript, we re-plated the mutants for Figure 2 (discussed in point 1) and observed a sensitive Δrad52 response to zeocin.

*9) The authors should reference a recent work from Song and co-workers (Kim et al., Sci. Rep. 2016) that also performed chemical crosslinking of yeast CAF-1, and that also came to the conclusion that the Cac1 subunit is the platform for interactions with other subunits. In addition, the crosslinking of not only individual residues but whole domains appears to differ in many ways between the two papers. For one example, the present work detected Cac1-histone crosslinks on the C-terminal half of Cac1, but many histone crosslinks map to the N-terminal half of Cac1 in the Kim et al. paper. Are there differences in the protocols used consistent with these observations? Does this change some of the conclusions here, and why or why not?*

We now reference this study and address these points here and in the manuscript. The manuscript by Kim, *et al.* (2016) reports some consistent cross-links with our own data. Specifically, we note similar cross-links between the middle domain (residues 118-334) of Cac1 to Cac2, Cac3, and histone H4. We also observe similar cross-links between the C-terminus (383-606) of Cac1 to Cac2 and histone H3. These observations support the similar conclusion that Cac1 scaffolds other proteins in the complex. Some differences exist between these two studies, however, which can be attributed to the following experimental and computational differences:

a) In our manuscript, we show in Figure 3—figure supplement 1 that Cac1 likely has folded domains around residues 118-334 and 460-606. These domains may extend even more, considering that only lysine-lysine, lysine-aspartate, and lysine-glutamate pairs are able to cross-link. Thus, some of the cross-links that appear to be spaced apart on a primary sequence may be consistent when considering the possible tertiary structure. For example, the observation of Cac1 K144 and K328 cross-linking to Cac2 K38 (Kim, *et al.* (2016), Figure 1) may suggest that two distinct regions of Cac1 interact with Cac2. Our intra-Cac1 cross-links in Figure 3—figure supplement 1 suggest that this is not the case, highlighting the value of our dataset.

b) The experimental procedures for Kim, *et al.* (2016) are not well-detailed in their manuscript. A reference to a prior methods paper is provided where several potential conditions are recommended. We believe that detailed reporting of the concentration or amount of proteins used, the reaction times of protein interactions and cross-linker interactions, the sample preparation protocol, mass spectrometry data acquisition, and data analysis parameters are all critical information for readers. We provided these details for the following essential reason.

We would like to emphasize that chemical cross-linking is not an equilibrium method and the number of cross-links observed only informs a researcher of proximal residues, but has little or no relation to the binding energy of protein-protein interactions. Rather, cross-linking is a capture method that generates information for more definitive protein-protein structural and/or biophysical experiments. This approach is extremely useful for architectural studies of complexes such as CAF-1-H3/H4, but should be complemented by extensive and careful dissection of the hypotheses generated. Thus, we employed equilibrium approaches, which validate our cross-linking findings. Specifically, we only observe cross-links between the C-terminal half of Cac1 to H3 and H4. This is consistent with our biophysical experiments, which show that full-length Cac1 binds to H3/H4 with a dissociation constant of 97 nM, while Cac1^386^ binds to H3/H4 with a K_Dapp_ of 210 nM (Figure 1 and Figure 4). The bulk of the Cac1-H3/H4 interactions that contribute to lower the free energy of a CAF-1-H3/H4 complex, therefore, indeed occur at the C-terminal half of Cac1.

Because cross-linking is a capture method, incubation times should always be reported. We have observed aggregation of CAF-1-H3/H4 complexes induced by cross-linkers when the samples are left for long times, and therefore optimized our procedure to allow protein-protein interactions to reach equilibrium and to saturate cross-linking efficiency, all in a time frame that minimizes protein aggregation. We are uncertain about incubation times used in the Kim, *et al.* (2016) paper.

Nevertheless, we feel that the results are largely consistent between the manuscripts. Kim, *et al.* (2016) observe many cross-links between the middle domain of Cac1 to H3 and H4. We also observe Cac1 K317 cross-linking to H4 E53. Thus, both studies support the interpretation that the Cac1 middle domain scaffolds interactions with histones. Kim, *et al.* (2016) observe one cross-link at the C-terminus of Cac1 to histones (Cac1 K444 to H3 K64), while we report many such cross-links. The results are consistent with the C-terminus of Cac1 interacting with histones. We have now discussed these comparisons in the Discussion section:

“Importantly, our cross-links are largely consistent with a recent report examining CAF-1 and CAF-1-H3/H4 cross-links (Kim et al., 2016). […] The EM data also supports the idea that Cac1 is a platform for these interactions, as Cac1 is physically associated with Cac2, Cac3, and H3/H4 (Kim et al., 2016).”

c) Finally, we use a different strategy for determination of cross-linked peptides than that used for in Kim, *et al.* (2016) They used the xQuest software, which analyzes a mix of unlabeled and isotopically labeled cross-linkers, and relies on false discovery rates (FDR) to inform statistical confidence of possible cross-linked peptides. This FDR approach suffers when working with a small sample size, which is the case with only CAF-1 and H3/H4. Furthermore, while isotopic cross-linkers help researchers obtain extra verification of cross-linked peptides, they provide unnecessary disadvantages. The spectra from an isotopic mixture has reduced signal intensity and makes searches more difficult. This discussion is expounded in Trnka, *et al.* (2014)

In contrast, we used the Batch-Tag Web function in Protein Prospector, which searches for a cross-linked peptide as a variable “modification.” This results in three separate scores based on y and b ion matches: an overall score for the cross-linked peptide, a score for one cross-linked peptide only, and a score for the other cross-linked peptide only. This level of analysis helps minimize false positives, as a higher percentage of mis-assignments consist of one high-scoring peptide linked to one poorer-scoring peptide, yet the overall score for the overall cross-linked pair is reported as high. Thus, knowing the score of the poorer-scoring peptide, as well as confirming the ion matches in the spectrum, are critical determinants for generating a high confidence assignment. Overall, we feel that our analytical approach is stringent. We have now discussed these comparisons in the Discussion section.

“We observe more histone cross-links to the Cac1 C-terminus whereas Kim, et al. observe more to the middle domain.

[…]

This approach compares favorably to approaches that do not show the score of the poorer-scoring peptide (Trnka et al., 2014).”

*Also, how do the EM analyses in the Kim paper comport with the model in Figure 6?*

We were excited to see that Cac1 appears to scaffold interactions with Cac2, Cac3, H3, and H4 from the cross-linking data in the Kim *et al.* study, which is consistent with our model. In addition, Cac2 and Cac3 do not appear to directly interact. However, the modest to low resolution cryo-EM data neither disproves nor improves our architectural model. This is because their EM structure is not of sufficiently high resolution to provide any detailed structural information: it looks like a kidney bean, which adopts a slightly altered shape with H3/H4 binding. Thus, we respectfully request to avoid having to speculate about this EM model in our text. Rather, we prefer to state:

“The cryo-EM data also support the idea that Cac1 is a platform for these interactions, as Cac1 is physically associated with Cac2, Cac3, and H3/H4 (Kim et al., 2016).”

*10) Subsection “Dynamic and Modular Architecture of CAF-1”: "N-terminal deletions of human p150 have detrimental effects on p150-mediated chromatin assembly and binding to p60 (Kaufman et al., 1995)." Don't the authors mean C-terminal deletions have these detrimental effects?*

Thank you for catching that mistake. We did mean “C-terminal deletions,” and have now made the correction.

*11) Third paragraph, subsection “Interactions of CAF-1 with H3/H4”: This paragraph is somewhat confusing. Why are the WH-histone tail crosslinks surprising? Just because this region also contacts DNA, why does that exclude it from also (or alternatively, in different molecules in a population) interacting with histones?*

We agree that this paragraph is confusing. We have pointed out that the C-terminus (ED to WH domains), especially the WH domain, has binding and conformational plasticity due to interactions with multiple binding partners. Thus, these cross-links should not be surprising. Since the plasticity of binding was discussed, we have removed the repetitive sentences.